# ONIX: a unified open-source platform for multimodal neural recording and perturbation during naturalistic behavior

Jonathan P. Newman[1,2,3,10], Jie Zhang[1,2,10], Aarón Cuevas-López [3,4,5,10],
Nicholas J. Miller[1,6], Takato Honda [1,2], Marie-Sophie H. van der Goes [1,6],
Alexandra H. Leighton [5], Filipe Carvalho[5], Gonçalo Lopes[7], Anna Lakunina [8],
Joshua H. Siegle [3,8], Mark T. Harnett[1,6], Matthew A. Wilson [1,2] &
Jakob Voigts [1,3,6,9] ✉

Behavioral neuroscience faces two conflicting demands: long-duration recordings from large neural populations and unimpeded animal behavior. To meet this challenge we developed ONIX, an open-source data acquisition system with high data throughput (2 GB s$^{-1}$) and low closed-loop latencies (<1 ms) that uses a 0.3-mm thin tether to minimize behavioral impact. Head position and rotation are tracked in three dimensions and used to drive active commutation without torque measurements. ONIX can acquire data from combinations of passive electrodes, Neuropixels probes, head-mounted microscopes, cameras, three-dimensional trackers and other data sources. We performed uninterrupted, long (~7 h) neural recordings in mice as they traversed complex three-dimensional terrain, and multiday sleep-tracking recordings (~55 h). ONIX enabled exploration with similar mobility as nonimplanted animals, in contrast to conventional tethered systems, which have restricted movement. By combining long recordings with full mobility, our technology will enable progress on questions that require high-quality neural recordings during ethologically grounded behaviors.

There is a growing recognition that, to maximize their explanatory power, neural recordings must be conducted during normal animal behavior. From the recent discovery that motor actions can dominate the activity of brain regions that were believed to be predominately sensory[1,2], to findings of different learning strategies between head-fixed and freely moving subjects[3], mounting evidence indicates that free behavior transforms the function of the nervous system. These observations are leading toward a consensus that learning[3,4],

social interactions[5,6], sensory processing[7,8] and cognition[9,10] are best addressed in animals that are engaged in naturalistic behavior.

In recent years, remarkable progress has been made on methods for tracking and quantifying animal behavior[11–42]. Parallel advances in recording technologies have enabled electrophysiology[43,44], optical imaging[45–47] and actuation of neural ensembles[48] in mobile animals. Still, applying these technologies, which are often bulky and require tethers, to study naturalistic behavior remains a major challenge. In

[1]Department of Brain and Cognitive Sciences, MIT, Cambridge, MA, USA. [2]The Picower Institute for Learning and Memory, MIT, Cambridge, MA, USA. [3]Open Ephys, Atlanta, GA, USA. [4]Department of Electrical Engineering, Polytechnic University of Valencia, Valencia, Spain. [5]Open Ephys Production Site, Lisbon, Portugal. [6]McGovern Institute for Brain Research, MIT, Cambridge, MA, USA. [7]NeuroGEARS, London, UK. [8]Allen Institute for Neural Dynamics, Seattle, WA, USA. [9]HHMI Janelia Research Campus, Ashburn, VA, USA. [10]These authors contributed equally: Jonathan P. Newman, Jie Zhang, Aarón Cuevas-López. ✉e-mail: voigtsj@janelia.hhmi.org

**Fig. 1 | ONIX, a unified open-source platform for unencumbered freely moving recordings. a**, Simplified block diagram of the ONI, illustrated via the tetrode headstage: multiple devices all communicate with the host PC over a single micro-coax cable via a serialization protocol, making it possible to design small multi-function headstages. **b**, The integrated nine-axis absolute orientation sensor and 3D tracking redundantly measure animal rotation, which drives the motorized commutator without the need to measure tether torque, enabling long recording durations. Small drive implants[44] enable low-profile implants (~20 mm total height). **c**, The ONIX micro-coax, a 0.31 mm thin tether, compared to standard 12-wire digital tethers. **d**, Torque exerted on an animal's head by tethers. Current tethers allow full mobility only in small arenas and in situations when the tether does not pull on the implant, while the ONIX micro-coax applies negligible torque. **e**, Performance of ONIX: with the 64-channel headstage, a 99.9% worst-case closed-loop latency, from neural voltage reading, to host PC, and back to the headstage (for example to trigger a light-emitting diode (LED)) of <1 ms can be achieved on Windows 10 (see also Extended Data Figs. 6 and 7). FPGA, field-programmable gate array; EIB, electrode interface board; FIFO, first-in first-out buffer.

larger animals like rats[49], primates[24] or even bats[50], wireless systems are available; however, in mice, which are the predominant animal model system in neuroscience, recordings are limited by the weight of recording devices. For example, a 6-g wireless logger can achieve only a 70-min long recording[51] and the weight of its batteries limits movement beyond slow locomotion, requiring that experiments be designed around the head torque imposed by the recording device[51]. Therefore, current technologies for mice and similar-sized species, do not allow for unencumbered motion, nor for recordings during behavior that unfolds over long periods or in large spaces, limiting our ability to capture neural activity during ethologically relevant behaviors.

## Results

To address this need, we developed an open-source multi-instrument hardware standard and application programming interface (API) (Open Neuro Interface (ONI); Extended Data Figs. 1 and 2). We then used ONI to implement a recording system called 'ONIX' – a modular and extendable data-acquisition and behavior-tracking system that greatly reduces the conflict between large-scale neural recordings and their impact on mouse behavior. The system uses a thin and light micro-coaxial (micro-coax) tether (center conductor and shield, easily replaceable, ~0.31 mm diameter and 0.37 g m⁻¹) compared to widely used options (for example, a custom multi-conductor of 3 mm diameter and 6.35 g m⁻¹; Fig. 1c), which causes minimal forces on the animal's head (Fig. 1d), together with a motorized commutator system that eliminates twisting of the tether, allowing long recordings (Figs. 2 and 3). The tether simultaneously powers and transmits data (150 MB s⁻¹, equivalent to 2,500 channels of spike-band electrophysiology data) to and from sensors and actuators. ONIX includes modular, miniaturized headstages (Figs. 1 and 4 and Extended Data Fig. 3) for passive electrical recording probes, tetrode drives[44] (via Intan RHD and RHS chips) and Neuropixels[43]

(1.0 and 2.0 probes and ball grid array (BGA)-packaged chips). In addition to neural activity, these headstages record a 6 degrees-of-freedom (d.f.) head pose at ~30 Hz via onboard sensors with ~2° angular and submillimeter position resolution (90% of jitter <0.02 mm in $x-y$ and <1 mm in $z$ at a 2-m distance; Extended Data Fig. 4) via an absolute angle sensor and a consumer-grade 3D-tracking system (Bosch BNO055 and HTC Vive). Real-time tracking permits the system to measure the rotation of an animal and automatically untwist the tether via a small motor (Fig. 1b and Extended Data Fig. 5) without requiring torque measurements. This approach removes the behavioral impact and time limits typically associated with tethered recordings and provides experimenters with the ability to monitor neural activity and behavior for arbitrarily long sessions in complex environments (Figs. 2 and 3).

ONIX is capable of submillisecond closed-loop neural stimulation on a standard (non-real-time) operating system (Fig. 1e and Extended Data Figs. 6 and 7). This level of latency, otherwise only achievable on specialized operating systems[52] or hardware[53], enables scientists to develop high-performance yet replicable closed-loop systems. Further, its hardware-agnostic, open-source C API allows scientists to more easily develop, share and replicate algorithms for such experiments in commonly used programming languages such as C++, C#, Bonsai[54] and Python, without being tied to specific hardware.

### Long-term neurophysiology with unimpeded behavior

To demonstrate ONIX's ability to perform rich, uninterrupted studies of freely moving mice, we performed ~8-h recordings while mice explored a 3D arena. The 1.5 × 1.5 × 0.5 m arena was constructed from hexagonal blocks of Styrofoam, cut to different heights, giving mice the opportunity to run, climb and jump (Figs. 2 and 3 and Supplementary Video 1). We exposed naive animals implanted with microdrives[44] to this unfamiliar environment without behavioral shaping or human intervention.

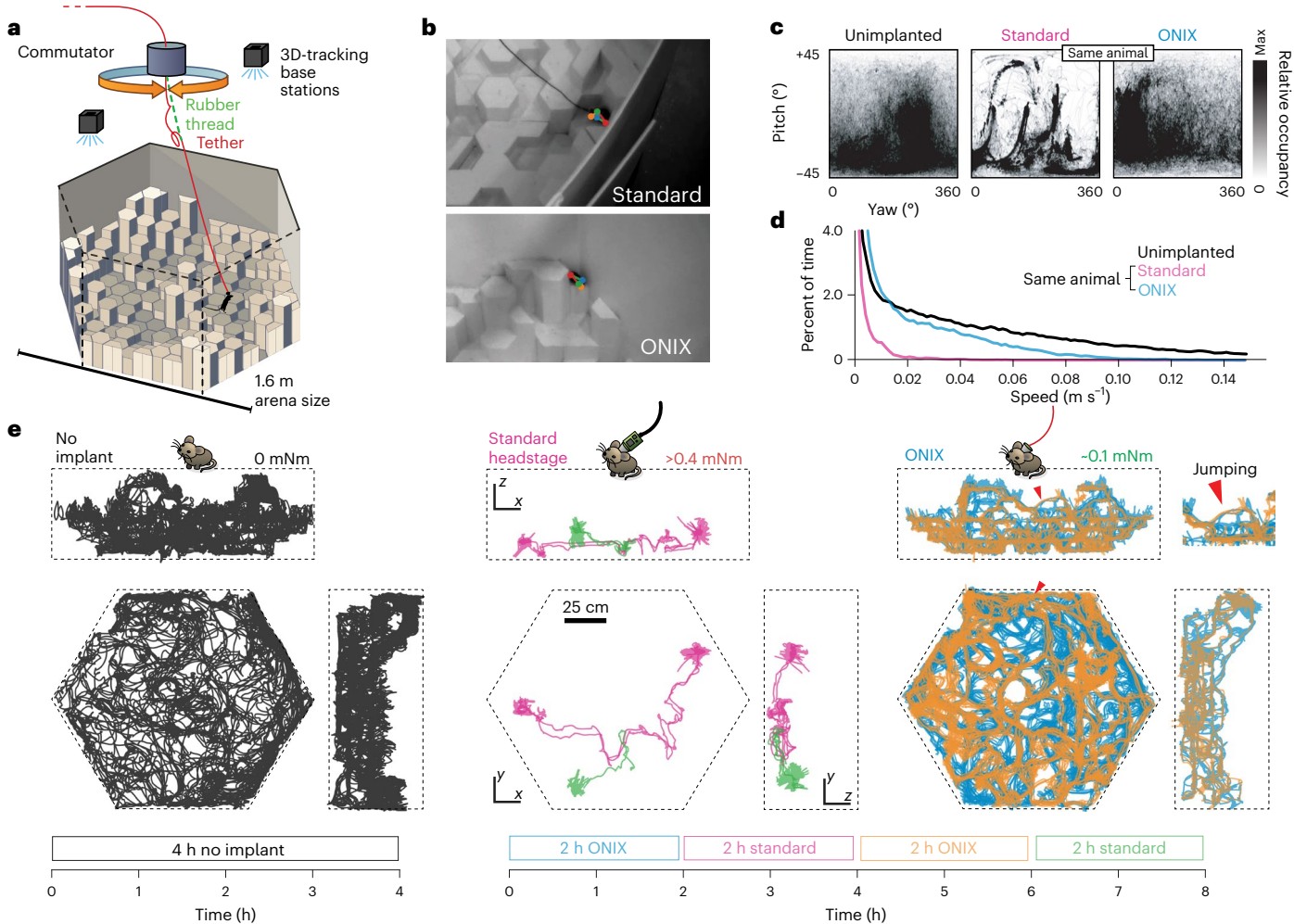

**Fig. 2 | Unrestricted naturalistic locomotion behavior with ONIX. a**, Overview of experiment. Mice were freely exploring a 3D arena made from Styrofoam pieces of varying heights. **b**, Unimplanted mice and mice with a standard tether (top) or ONIX micro-coax (bottom) were tracked in 3D using multicamera, markerless pose estimation[31]. **c**, Head yaw and pitch occupancies over the course of a recording. **d**, Speed distributions over the course of a recording. **e**, Two-dimensional projection of mouse trajectories over the course of a recording session.

We compared the mouse behavior achievable with the ONIX system to a typical, modern acquisition system (Fig. 2c–e), by attaching a standard tether (Intan 'SPI cable', 1.8 mm in diameter), counterweighted with an elastic band to eliminate the tether weight, alongside the micro-tether. This allowed use of the ONIX headstage for position measurements and commutation/untwisting, effectively adding torque-free commutation to the SPI tether, while imposing the mechanical effect of the weight of a traditional tether on the mouse. The tethers were switched every 2 h over an 8-h recording session (Fig. 2e). Except for tether exchanges, no experimenter was present. Even with our zero-torque commutation, the additional forces imposed by the standard tether (head torque >0.4 mNm, measured in separate experiment; Fig. 1d) had a substantial deleterious effect on exploratory behavior and freedom of head movement (Fig. 2c–e; quantified via entropy of the spatial occupancy and head position distributions, median entropy 4.21 versus 0.287 bits for spatial occupancy, and 0.75 versus 0.50 bits for heading; $P < 0.0001$, Wilcoxon rank-sum test). When only the micro-coax tether was used (head torque ≈0.1 mNm), the animal resumed free exploration of the arena (Fig. 2e). Some digital tethers, such as the twisted pairs used with Neuropixels[43] or coaxial cables used in Miniscope implants[45] (~1.3 mm OD), would fall in between the standard SPI cables used here and the ONIX micro-coax in terms of weight and flexibility, but they would lack zero-torque commutation, impacting behavior and limiting recording duration.

To compare the behavior achieved with ONIX to an implant-free condition, we used five synchronized cameras for markerless 3D tracking[31] of nonimplanted mice (Fig. 2e). The degree of arena exploration and the head orientation distributions of ONIX versus nonimplanted mice were statistically indistinguishable with 4 h of data per condition yielding largely overlapping confidence intervals (one mouse per experiment, quantified via entropy of the occupancy distributions, 95% confidence intervals for spatial occupancy: 0.208–0.369 for nonimplanted versus 0.224–0.383 for ONIX, and confidence intervals for heading: 0.479–0.536 for nonimplanted versus 0.474–0.533 for ONIX). The median and maximum running speed of the implanted mice was reduced by a factor of ~2 compared to animals with no implant; however, the ONIX micro-coax provided ~12× increase in median running speed and ~2× maximum speed compared to the standard tether (Fig. 2d).

To demonstrate the utility of long recordings without behavioral disruption, we conducted a 7.3-h recording with a tetrode drive implant[44] in the retrosplenial cortex in the 3D arena (Fig. 3). Mice spontaneously jumped to heights of >10 cm (Fig. 3a,b), allowing us to observe neural activity during jumps (Fig. 3f). This behavior was absent in mice with the heavier tether. We did not observe increased brain motion during high-speed motion or jumps (Extended Data Fig. 8). To demonstrate the ability of the system to perform very long, uninterrupted recordings without experimenter supervision (outside of visual

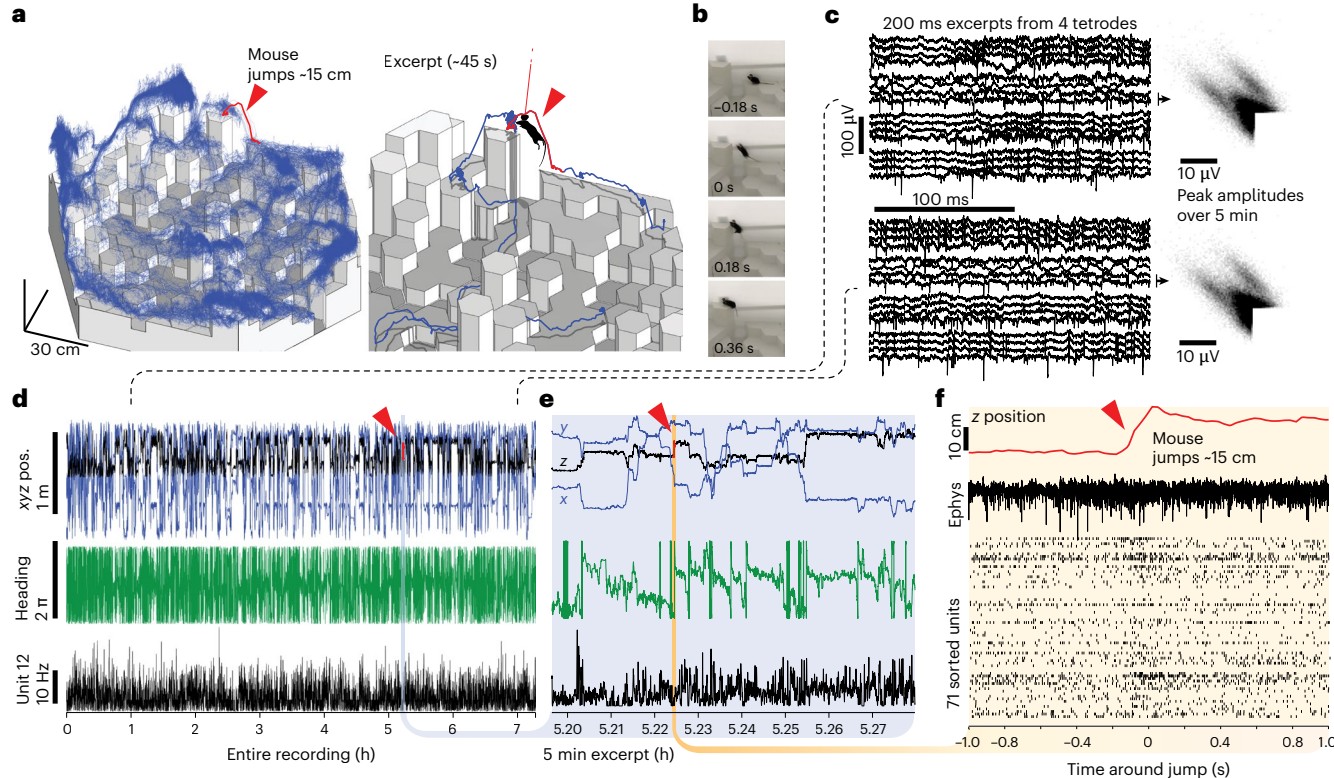

**Fig. 3 | Stable long-term recordings during naturalistic locomotion.**
**a**, Position of one 3D-tracking sensor on the headstage during a 7.3-h-long ONIX recording during which the mouse was free to explore the 3D arena. Red trace and excerpt show one of multiple instances of the mouse spontaneously jumping from a lower to a higher tile. **b**, Video frames of the jump (the tether is too thin to be visible at this magnification), see Supplementary Video 1. **c**, Raw voltages and spike peak amplitudes from two channels at hour 1 (top) and hour 7 (bottom) of the recording. **d**, 3D position, heading and smoothed firing rate of entire recording. **e**, Same data as in **d**, for excerpt around jump. **f**, z-position, raw voltage trace example and sorted spikes from 71 neurons during the jump.

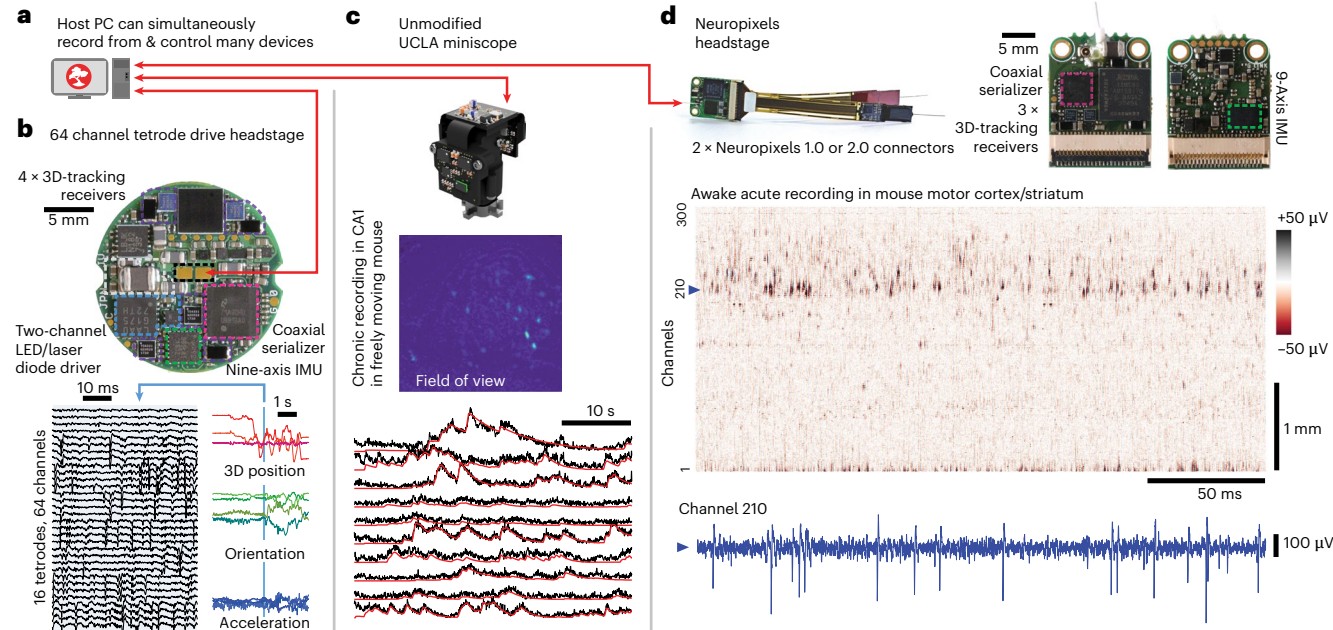

**Fig. 4 | ONIX is compatible with existing and future recording technologies.**
**a**, ONIX, together with Bonsai, can simultaneously record from and synchronize multiple data sources, such as tetrode headstages, Neuropixels headstages and/ or UCLA Miniscopes. **b**, 64-channel extracellular headstage, as used in Figs. 1–3, with 3D tracking, electrical stimulator (Extended Data Fig. 10), dual-channel LED driver and inertial measurement unit (IMU) (bottom side; not shown) (top). Example neural recording and corresponding 3D-pose traces collected from the headstage (bottom). **c**, ONIX is compatible with existing UCLA Miniscopes (v.3 and 4)[45,55]. Maximum projection after background removal of an example recording in mouse CA1 (middle). Background-corrected fluorescence traces (black) and CNMF output (via Minian[63], red) of ten example neurons (bottom). **d**, An ONIX headstage for use with two Neuropixels probes and IMU to enable torque-free commutator use for long-term freely behaving recordings. A voltage heat map shows a segment from a head-fixed recording. A voltage time series from the channel indicated by the dotted line is shown in blue.

health and safety checks), we recorded local field potentials in a mouse in a large home cage for 55 h (Extended Data Fig. 9). This shows that the system can keep the tether tangle-free for virtually unlimited periods and is reliable enough to allow for multiday recordings.

## An open standard for data acquisition systems

Finally, we demonstrate the ONI standard's flexibility by using the same ONIX system to acquire from and control two additional widely used, third-party devices: UCLA Miniscopes[45,55] (Fig. 4c and Extended Data Figs. 5 and 8) and Neuropixels probes[43] (Fig. 4d). These recordings can also be performed simultaneously if needed. By using the Bonsai software[54] for data acquisition, we also demonstrate integration of synchronized multicamera tracking (Fig. 2 and Extended Data Fig. 5). Bonsai enables the integration of real-time processing tools such as animal tracking via real-time DLC[56] or SLEAP[34], enabling experiments that react to animal behavior with high precision.

For developers, the ONI hardware standard and API streamlines the development of new probe and sensor technologies into headstages that have immediate integration with existing technologies, which lowers the barrier for individual laboratories to create custom instruments designed for specific experiments. Similarly, ONI simplifies the development of new data acquisition systems, for example ultrafast camera systems[57], by providing a scalable easy-to-use interface for communication between software and field-programmable gate array (FPGA) firmware, and ensures interoperability between these systems. Parts lists and design documents for the system are available at https://github.com/open-ephys (see the 'Code and design file availability' section for details).

## Discussion

Our system provides a probe-agnostic, open-source interface for use in neuroscience. It allows long and high-bandwidth recordings in mice and similarly sized animals during naturalistic behaviors comparable to those of nonimplanted animals. This ability will accelerate progress in many areas of research that currently rely on limited behavior in small boxes or over short timespans or with limited neural data or behavioral freedom, such as motor learning[58], sensory processing during natural behaviors[8], social behaviors[59] (with recordings from one animal), play[9] or on cognitive aspects of spatial behaviors[3,60–62] by allowing unimpaired motor behavior, reducing animal fatigue over time and enabling navigation in large environments.

## Online content

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

## Methods

In-depth and up-to-date documentation of the hardware, software and detailed user guides is found at https://github.com/open-ephys and documentation on the ONIX system is at https://open-ephys.github.io/onix-docs.

### Design goals and high-level architecture

Neuroscience data acquisition systems often use proprietary and/or single-purpose interfaces and protocols for device communication in the name of performance and commercial advantage[64]. Some side effects of this situation are device lock-in and difficulty of device interoperability. On the other hand, while open-source hardware, including our own designs, provides access to communication protocols and hardware, these protocols and physical interfaces generally underperform commercial options and tend be nonstandardized and brittle, making them difficult to extend and maintain.

Supplementary Table 1 summarizes physical interfaces, firmware, communication protocols and host APIs used by several open-source hardware projects in widespread use for systems neuroscience research. Each of these projects employs a different set of interconnects, device drivers, firmware and APIs. Their designs are tailored for the features of a particular sensor and are difficult to extend to general-purpose data acquisition. Further, each of these solutions requires one-to-one scaling of headstage tether conductors with the number of sensors at the headstage and precludes the combination of high-throughput recordings with naturalistic behavior.

To address these limitations, we have taken a top-down approach to design communication protocols, firmware and host API to support acquisition from any mixture of devices (Figs. 1a and 4). Specifically, our designs meet the following requirements:

(1) Heterogeneous device acquisition and control
   (a) Well-defined communication protocols that support bidirectional communication with any mixture of sensors or actuators.
   (b) Generic, low-level API that takes inspiration from existing, widely used libraries and is focused on the creation of high-level language bindings.
   (c) Bus-agnostic: firmware, protocols and API should not rely on a particular physical layer between headstage and host computer (for example wireless or tethered) or hardware interface and CPU (for example USB, Ethernet or PCIe).
(2) Closed-loop performance
   (a) Submillisecond headstage to host PC round-trip time.
   (b) Ability to drive neural stimulation devices directly from headstage.
   (c) First-class bidirectional communication with host computer.
(3) Practical in vivo use
   (a) On-headstage processing.
   (b) Generic data serialization protocol and firmware can be used with single-wire tether or wireless transmission.
   (c) Physical headstage connectorization bounded to a single coaxial connection regardless of the number of devices on headstage.
   (d) Bandwidth for up to 1,000 s of recording channels sampled at 30 kHz.
(4) Use standard parts and interfaces
   (a) Make use of standard parts created for large markets (mobile computing, automotive, etc.).
   (b) Use standard and widely used connectorization and avoid specialty hardware wherever possible.
   (c) Use standard programming practices that emulate other widely used and easily extended open-source APIs for hardware interfacing and require minimal dependencies.
   (d) Cross-platform.
   (e) Minimize cost.

The resulting system consists of five major components (Extended Data Fig. 1). Reusable headstage firmware modules (1) allow headstages to interface with an arbitrary arrangement of sensors and actuators. Device data streams are combined and packaged at the headstage using generic and extensible firmware modules and a well-defined serialization protocol (2) over a micro-coax cable (serialization protocol is interconnect-agnostic and could use a wireless link). Data are deserialized by the firmware running on an FPGA interface board inside the host computer (3). A device driver is dynamically loaded by the API to match with firmware running on arbitrary deserialization hardware to pass data to the user-space memory using a high-speed communication protocol (4). Any peripheral bus capable of realizing this protocol is supported, but we focus on the use of the PCIe bus due to its high performance. Finally, a C API (5) provides software access to the resultant data and control streams. This API's simplicity makes it usable for arbitrary data streams and amenable to the creation of language bindings for easy integration with existing acquisition software. Our implementation of this architecture greatly decreases and stabilizes closed-loop response latencies compared to existing open-source systems and permits scalable combination of miniature head-borne sensors and actuators to simplify complex closed-loop experimentation.

### Open Neuro Interface: a standard for head-borne instrumentation

This work was built on the ONI specification, a set of general-purpose communication protocols, device driver specifications and programming interfaces to support arbitrary mixtures of hardware. The ONIX system outlined here is one implementation of the ONI specification.

ONI is an open standard describing a high-speed interface between a computer and a collection of devices, which can be of a different nature. Its goal is to provide a single, unified protocol to communicate with the variety of instruments widely used in neuroscience such as electrophysiology acquisition devices, tracking systems, cameras or stimulators. It defines both a general-purpose communication protocol, along with an API specification. It does not require a specific physical layer, leaving that open to different implementations, but only states how the data should be organized, how different communication channels should behave from the perspective of the host computer and the nature of a compliant API implementation. The specification itself can be found at https://github.com/open-ephys/ONI. Extended Data Fig. 2 provides a simplified overview of the structure of the standard.

### Acquisition performance

Performance of any acquisition system is measured in bandwidth, indicating the amount of data it can acquire per second, and latency, which measures the time between an event and the ability to act on it. The maximum bandwidth allowed by the x4 PCIe Gen2 interface is 2 GB s$^{-1}$; however, the current ONIX system operates at a maximum theoretical bandwidth of 1.6 GB s$^{-1}$ total. Data transfer between the hardware and host application does not occur continuously, but in blocks, with each individual transfer operation incurring a small overhead. One feature of the ONI specification is that this block size must be settable to allow the user to balance real-time responsivity against overall bandwidth. Large block sizes increase bandwidth by reducing overheads and system call frequency at the cost of increased data latency. Extended Data Fig. 6 shows measured bandwidth using a load-testing device in the host firmware.

While latency, measured in time, is related to block size, its actual impact is dependent on the data origin. This effect is more detrimental for devices generating small packets at high frequencies (MHz), which are rarely the case in neuroscience, as opposed to larger sample sizes at the kHz range. For example, a 8,192-byte block size would introduce a latency of 1,024 samples on a simple, eight-byte time-stamping device;

however, for the case of four Neuropixel probes, with a sample size of 480 bytes each, an 8,192-byte block transfer would imply a latency of fewer than five samples. Although total latency is dependent on block size, there is a fixed, minimum latency associated with the processes of transfer initiation. For the ONIX system, transmission latency was measured with the smallest block size and a simple C program responding to a digital event. Under these conditions, 150 μs of maximum transmission latency were measured. Figure 1e shows closed-loop latencies measured for a round-trip from electrophysiology data to optical or electrical stimulation on the headstage with a 64-channel Intan headstage. In general, response times will be highly dependent on the hardware producing data, the physical layer between the acquisition system and the host computer, the host computer itself, and most notably, complexity of the real-time algorithm acting on the data. Because ONI forces the block size to be tunable, it can be adjusted to empirically optimize response times for a particular setup to compensate for these factors.

## System reliability
In beta-testing the system over the last 3 years, the main beta-testers of the mouse 64-channel headstages in the Harnett laboratory accumulated over 800 h of total recording time across this study, with two studies using smaller arenas[61,65] and various test recordings. During this time, we improved various aspects of the system to improve reliability, for example moving components away from areas where they could become damaged over time due to handling. No spontaneous failures due to acquisition system hardware instability were observed during experimental recordings. At various times we encountered software issues such as camera driver instability, operating system crashes or memory leaks causing stability issues for long recordings. The bugs related to the ONIX system were resolved during this beta-test period, and recordings of virtually unlimited length, such as the 55-h experiment (yielding ~711 GB of electrophysiology data) shown in Extended Data Fig. 8 are possible.

All aborted recordings due to hardware failure were due to breaks in the data and power connection on the tether from cases where mice managed to grab and bite the tether. This occurred either due to user error in the counterbalancing (allowing too much slack so that mice could grab the tether) or due to early design errors in the enclosure, leading to edges in the enclosure that the tether could snag on, again allowing the mice to bite it. In these cases, the tether could be replaced quickly, the two-conductor coax is not particularly expensive and is easy enough to solder even for relative novices. Headstage connectors that are expected to undergo many connection cycles are using high-cycle-count-rated industry standard parts that have a proven track record of long-term reliability in neuroscience, and we designed the headstages so that the part of the connector that experiences wear is on the animal (and will therefore be replaced with each new implant) and the part that does not experience wear is on the headstages.

## Headstages
Using the architecture detailed above, headstages for extracellular electrophysiology, using amplifier/digitizer chips by Intan Technologies (https://intantech.com/), as well as headstages for use with dual Neuropixels[43,66] probes (probe v.1.0 and 2.0), headstages supporting electrical stimulation on every Ephys channel, compressive sensing cameras[57] and others were designed. Images of some of these headstages that were used in the paper and listings of major features are provided in Fig. 4. In the following text we provide detailed descriptions of each of these features.

## Size and weight
The Intan headstage is designed to be mounted flat on top of the EIB of an tetrode microdrive[44]. This parallel, rather than the traditional orthogonal, mounting scheme was chosen to reduce torque on the

animals' head during freely moving behavior (Extended Data Fig. 3). Headstage-64 is 19 mm in diameter. When the printed circuit board is completely populated it weighs ~0.95 g. The Neuropixels headstage was designed as a traditional dual-probe layout and weighs ~1.1 g without the probes attached.

## Headstage input–output, serialization and physical interface
A key feature of the firmware provided with the project is the ability to serialize an arbitrary set of asynchronous data sources into a single data stream. This stream can then be transmitted back to the host using any transceiver with appropriate bandwidth. Our choice of transceiver between our headstage and the host PC was inspired by the UCLA Miniscope[45], which uses a Texas Instruments DS90UB933A/34A serializer–deserializer (SERDES) pair to combine power, high-bandwidth data transmission and low-bandwidth headstage control into a single micro-coax cable. Although our headstages focus on the use of this transceiver, it is important to note that the modular design of our firmware allows any appropriate link to be used. For example, other existing ONIX headstages that were not used in this paper use DS90UB9353A/354A SERDES, which provides a mobile industry processor interface, increased bandwidth and a more responsive real-time control. Further, wireless communication, for example via wifi or Bluetooth, could also be used to enable real-time data streaming using an appropriate chipset on the headstage with minimal changes required in firmware.

In the tethered configuration, the coaxial cable is the only external connection to the headstage. Power (DC), a control 'back-channel' (70 MHz) and a data 'forward-channel' (700 MHz) occupy different portions of the RF spectrum and therefore can be resolved as distinct signal streams. Power is DC-coupled to of the coaxial interface via an inductive path, while a capacitive path from the serializer is used to AC couple in the control and data signals. The reverse occurs at the deserializer. Although this chip is intended for use with camera sensors, it can be repurposed to transmit arbitrary data using the headstage and host FPGAs to provide a camera-like digital interface for arbitrary data. We do this by repurposing the SERDES data and control interface using an FPGA to spoof a camera and camera decoder at serialization and deserialization ends, respectively.

## Headstage FPGA
The 64-channel and Neuropixels headstages use the Intel MAX10 FPGAs for peripheral device control (for example, stimulation timing), sensor sample collection, data packaging and buffering, and serializer interfacing on the headstage. This FPGA was chosen due to its small size, integrated flash storage, phase-locked loops, and 32-bit soft processor. The 64-channel headstage uses an 81-pin wafer-level chip-scale package (4×4 mm footprint; Figs. 1a and 4b). The choice of FPGA is not critical as long as the device is physically small and has a low power budget. For instance, we have used Lattice Crosslink devices for other headstages not used or described in this paper.

## Headstage sensors and actuators
**Electrophysiology.** The Intan headstage performs multichannel electrophysiology using a BGA-packaged Intan RHD2064 64-channel bioamplifier/digitizer chip. The ONI-compliant firmware and API permit read and write access to the RHD2064's 64 control registers from the host PC. The Neuropixels headstage interfaces with the Neuropixels chip using a similar MAX10 FPGA to the 64-channel headstage. It shares other digital logic blocks for 3D pose tracking with the 64-channel Intan headstage but omits the optogenetic and electrical stimulation circuits.

## Electrical microstimulator
The Intan headstage features a single constant-current, bipolar, electrical stimulation circuit (Extended Data Fig. 10). Connections to the stimulation circuit are routed through the high-density connectors on the bottom of the headstage to the EIB where static or movable stimulation

electrodes can be attached using standard methods. The stimulator is an improved Howland current pump with a bipolar 15 V supply. The stimulation current is measured on the headstage and routed to an auxiliary input of the Intan chip. Outside of the stimulus pulses, the circuit provides charge balancing by shorting the stimulation electrode to ground. Component values have been chosen to optimize circuit stability over a wide range of electrode impedances. The operation remains stable for macroelectrodes (for example low-impedance, cut, stainless-steel wire and microelectrodes up to 1 MΩ at 1 kHz). The circuits can produce up to 1.5 mA of bipolar current within the bounds of its ±15 V compliance voltage range. Although this circuit consists of multiple components, the firmware and API provide an abstract control interface that allows high-level configuration and stimulus timing of the entire circuit. In this context, its operation is very similar to a Master8 or PulsePal[67].

### LED/laser diode driver

The Intan headstage provides two high-current LED/laser diode drivers for optogenetic stimulation (On Semi CAT4016). The maximal current is set over a wide (-10 mA to 800 mA) range via an external digital potentiometer. The optical power can then be adjusted linearly and synchronously across all channels within this range over eight levels per diode load. Like the electrical stimulator, this subcircuit is controlled as a single device with the API using parameters similar to a Master8 or PulsePal[67].

### 3D tracking system

Both the Intan headstage and the Neuropixels headstage provide a set of sensors for precise, room-scale 6 d.f. head tracking. An integrated nine-axis IMU provides kHz-scale measurements of head pose and angular acceleration. The 64-channel headstage uses a Bosch BNO55, which provides low-frequency compass data to determine an absolute bearing. A select set of IMU control registers are exposed through the firmware and API for calibration and to adjust data output type. For instance, the BNO55 allows on-chip sensor fusion to directly report the pose in some acquisition modes. This fusion mode can be used to drive the motorized commutator and keep the tether from twisting when the animal rotates.

In addition to the IMU, each headstage has multiple light-to-digital transceivers to capture laser sweeps from 'lighthouse' tracking stations. Fusion of IMU data with laser sweep timing information can be used to robustly deduce millimeter level 3D position within a -10 m³ environment (Extended Data Fig. 4 shows the precision of only the optical tracking method). Each of these devices is treated as a separate data source by the API and raw data are streamed to the host computer for each sensor for sensor fusion. In the future, it is conceivable that these operations could be moved to the headstage FPGAs embedded softcore processor and the entire tracking system treated as a single abstract device that sends precalculated 6 d.f. pose information.

### Control board

Following the ONI specification, the ONIX control board (Fig.1a) was designed to aggregate data from different hubs and devices and interface with the computer through a high-speed PCIe bus. The Numato Nereid board with a Kintex-7 FPGA was used as the base of the system. An ANSI/VITA 57.1 standard FPGA Mezzanine Card (FMC) daughter card was made containing all custom electronics for interfacing with external devices. In the future, control boards with other interfaces such as a USB could be developed that will function interchangeably, though might strike a different balance between ease of use, throughput and closed-loop latency.

The main components of the FMC board are two DS90UB9334A deserializers that communicate through a coaxial link with different hubs. External connection to the deserializers is made via MMCX connectors. These devices require two different power inputs: 3.3 V for input–output lines to the FPGA and 1.8 V for the core. Those are provided through 3.3-V pins in the FMC connector, originating in the

Nereid board, and a DC–DC Buck converter to efficiently derive the 1.8 V lines. Power for the external devices, transmitted through coaxial cables, is derived from a 12 V source also provided through the FMC connector. Instead of being converted to a fixed voltage, a combination of configurable step-down converters and digitally controlled potentiometers was used to control the link voltage for each port. This can compensate for larger cables that cause higher voltage drop or adjust for different devices. The control board also contains interfaces for general purpose analog and digital signals. It features 12 analog lines that can be, through digitally controlled analog switches, independently routed to either a multichannel ADC or DAC. The board does not feature direct digital lines, however, but five high-speed differential pairs (two output and three input), which can be used to interface with a variety of digital systems. A breakout board can be attached to this interface to expose this bus as 8 bits of digital input and 8 bits of digital output. For additional synchronization capabilities, the control board contains two buffered high-speed clock inputs and one clock output, accessible through coaxial MMCX connectors. It also features an internal connector with four differential pairs of configurable direction, designed to connect multiple boards to work in a synchronized manner.

### Support hardware

**Active motorized cable commutator.** The commutator consists of a commercial RF rotary joint with a top (static) and a bottom (rotating) SMA connector. The bottom connector is actively rotated by a stepper motor via a pair of custom 3D-printed gears, one of which is attached to the slip ring and one to the motor axle. A custom driver board interfaces with the host PC with a standard USB interface and generates appropriate motor control signals as instructed by the host PC. Capacitive touch buttons on the outer side of the printed circuit board allow for manual operation of the commutator. The entire system is powered by the 5 V provided from the USB interface.

### Electrode interface boards

To support the use of the headstages, we created EIBs adapted for interfacing with tetrodes drives[44] or other electrode or drive implants that use Omnetics connectors. The designs primarily cater to tetrode use, but the schematics can be easily adapted to different PCB form factors for other probe designs. In addition to recording electrodes, all EIBs include outputs from the headstage for onboard LED drivers and electrical microstimulators, which can be connected to drivable LED pigtails or electrodes, respectively. Electrodes can be soldered to through-holes or compression-connected using conical, gold-plated pins (available from NeuraLynx). These boards contain four copper layers. Sensitive analog traces are routed in internal layers. The top and bottom layers are connected to the system ground providing near-complete Faraday shielding of analog traces.

### Mouse behavior and neural recordings

All mice used for chronic electrophysiology verification were used for a separate study[61] before use in the experiments described here. Nonimplanted mice used for behavior verification were handled under the same protocol as outlined here, but with no surgical intervention.

**Drive implants.** Lightweight drive implants with 16 movable tetrodes were built as described previously[44,61]. The tetrodes were arranged in an elongated array of approximately 1,250 × 750 μm, with an average distance between electrodes was 250 μm. Tetrodes were constructed from 12.7 μm nichrome wire (Sandvik Kanthal, QH PAC polyimide-coated) with an automated tetrode twisting machine[68] and gold-electroplated to an impedance of approximately 300 kΩ.

**Surgery.** Male and female mice (C57BL/6 RRID: IMSR_JAX:000664) were aged 8–15 weeks at the time of surgery. Animals were housed in pairs or triples when possible and maintained on a 12-h cycle.

All experiments were conducted in accordance with the National Institutes of Health (NIH) guidelines and with the approval of the Committee on Animal Care at the Massachusetts Institute of Technology (MIT). All surgeries were performed under aseptic conditions under stereotaxic guidance. Mice were anesthetized with isoflurane (2% induction, 0.75–1.25% maintenance in 1 l min$^{-1}$ oxygen) and secured in a stereotaxic apparatus. A heating pad was used to maintain body temperature and additional heating was provided until the mice were fully recovered. The scalp was shaved, wiped with hair-removal cream and cleaned with iodine solution and alcohol. After intraperitoneal injection of dexamethasone (4 mg kg$^{-1}$), carprofen (5 mg kg$^{-1}$), subcutaneous injection of slow-release buprenorphine (0.5 mg kg$^{-1}$) and local application of lidocaine, the skull was exposed. The skull was cleaned with ethanol and a thin base of adhesive cement (C&B Metabond and Ivoclar Vivadent Tetric EvoFlow) was applied. A stainless-steel screw was implanted superficially anterior of bregma to serve as the electrical ground. A 3-mm craniotomy was drilled over central midline cortex, a durotomy was performed on one side of the central sinus and tetrode drives[44] were implanted above the retrosplenial cortex, at around AP −1.25 to −2.5 mm and ML 0.5 mm, with the long axis of the tetrode array oriented AP, and the tetrode array tilted inward at an angle of ~15–20° and fixed with dental cement. The ground connection on the drive was connected to the ground screw, and the skin around the drive implant was brought over the base layer of adhesive as much as possible to minimize the resulting wound margin, sutured and secured with surgical adhesive.

At the time of implant surgery, only two of the tetrodes were extended from the drive to serve as guides during the procedure. All other tetrodes were lowered into superficial layers of cortex within 3 days after surgery. Mice were given 1 week to recover before the start of recordings.

**Chronic electrophysiology.** After implant surgery, individual tetrodes were lowered over the course of multiple days until a depth corresponding to cortical layer 5 was reached and spiking activity was evident. Data were acquired with an Open Ephys[69] ONIX prototype system at 30 kHz using the Bonsai software[54]. The tether connecting the mouse headstage to the acquisition system was routed through a motorized commutator above the arena and was counterbalanced via a segment of flexible rubber tread.

**Spike sorting.** Voltage data from the 16 tetrodes, sampled at 30 kHz were bandpass filtered at 300–6,000 Hz, and a median of the voltage across all channels that were well connected to tetrode contacts was subtracted from each channel to reduce common-mode noise such as licking artifacts.

Spike sorting was then performed per tetrode using Mountainsort[70] (https://github.com/Flatironinstitute/mountainsort_examples) and neurons were included for further analysis if they had a noise overlap score <0.05, an isolation score >0.75 (provided by Mountainsort[70]), a clear refractory period (to ensure spikes originated from single neurons) and a spike waveform with one peak and a clear asymmetry (to exclude recordings from passing axon segments), and a smooth voltage waveform and histogram (to exclude occasional spike candidates driven by electrical noise). Units were not excluded based on firing rates, tuning or any higher-order firing properties.

**Miniscope recording.** Basic surgery procedures were the same as described for drive implant surgeries. Mice were then unilaterally microinjected with 500 nl of AAV1-syn-jGCaMP7f-WPRE12 (Addgene) at 50 nl min$^{-1}$ using the stereotactic coordinates: −2.1 mm posterior to bregma, 1.5 mm lateral to midline and −1.5 mm ventral to skull surface. Two weeks later, a gradient refractive index lens (GRIN) was implanted above the previous injection site. A 1.5-mm diameter circular craniotomy was centered at the previous virus injection site. Artificial cerebrospinal fluid was repeatedly applied to the exposed tissue to prevent drying. The cortex directly below the craniotomy was aspirated with a 27-gauge

blunt syringe needle attached to a vacuum pump. The GRIN lens (1.0 mm diameter, 0.5 pitch and 4.0 mm length; Inscopix) was slowly lowered with a stereotaxic arm above CA1 to a depth of 1.45 mm ventral to the surface of the skull. The GRIN lens was then fixed to the skull using cyanoacrylate glue and dental cement. Two weeks later, a small rectangular baseplate was cemented onto the animal's head atop the previously formed dental cement. During imaging, the microendoscope (UCLA Miniscope v.4) was fixed in place inside the baseplate. The microscope's focus was adjusted electronically before recording to ensure the cells were in focus. For analyses of brain motion (Extended Data Fig. 8), a Miniscope implant was carried out as described in accordance with the NIH guidelines and with the approval of the Committee on Animal Care at HHMI Janelia Research Campus. The position of the Miniscope was tracked using SLEAP[34], and brain motion was measured by computing the motion of the Miniscope image using an fft-based image stabilization algorithm[63].

**Neuropixels recording.** To validate the use of ONIX with Neuropixels probes, a separate experiment was carried out at the Allen Institute following protocols approved by the internal Institutional Animal Care and Use Committee under an assurance with the NIH Office of Laboratory Animal Welfare. A ChAT-IRES-Cre transgenic mouse (Jackson Labs) was implanted with a titanium headframe, and most of the left parietal skull plate was removed and replaced with a 3D-printed cap. After 4 weeks of recovery, the protective silicone elastomer was removed from the skull cap under isoflurane anesthesia and replaced with a 1-mm thick layer of Duragel (DOW DOWSIL 3-4680). In the same procedure, a silver ground wire was inserted through one of the anterior holes in the 3D-printed skull cap until it was just touching the brain surface. Starting on the following day, the mouse was habituated to head fixation on the recording rig for three consecutive days. On the day of the recording, a Neuropixels 1.0 probe connected to an ONIX headstage was inserted through one of the holes in the skull cap at a rate of 200 μm min$^{-1}$ to a depth of 2.5 mm. After waiting 5 min for the probe to settle, 384 channels of action potential band and local field potential band data were recorded for 15 min at 30 kHz using Bonsai. The recording was made in external reference mode, with the Neuropixels ground and reference soldered together. Before visualization, the raw action potential band data were phase shifted and high-pass filtered (300 Hz cutoff) and the median was subtracted using SpikeInterface[71].

**Long-term recording.** To validate the long-term stability of the recording system and the ability to correct for mouse rotation with no residual drift, a separate experiment was carried out at HHMI Janelia Research Campus approved by the Janelia Institutional Animal Care and Use Committee and in compliance with the standards set forth by the Association for Assessment and Accreditation of Laboratory Animal Care. A male PWK/PhJ × C57BL/6 (Jackson Labs) F1-cross mouse was implanted with a laminar probe (Cambridge Neurotech) in the prefrontal cortex following the same procedure as outlined for tetrode implants. After recovery and use in a separate experiment, the animal was transferred to a large rat cage. Neural data were acquired in a mouse holding room with an automated 12-h light–dark cycle without interruptions or experimenter intervention (outside of a daily health check) and post-processed in the same way as for the other experiments. Animals were remotely checked every 5 h to ensure that no tether tangling was evident, but no twisting was observed and no intervention was necessary. Neural data from one channel was bandpass filtered in the 6–10 and 30–50 Hz bands to compute spectral power across awake and sleep phases. Behavioral activity levels were quantified by smoothing the norm of the 3D acceleration vector from the headstage at 1 Hz.

**Behavioral experiment hardware.** Behavior experiments were carried out in a hexagonal arena of 1.6 m diameter. Individual floor tiles varied in height in a randomly chosen pattern to give mice the ability to run and

jump across gaps spontaneously. The floor tiles were made of Styrofoam and painted with a water-resistant acrylic primer. All behavioral experiments were conducted in Bonsai[54], the code for conducting the recordings is available on the ONIX GitHub repository (https://open-ephys.github.io/onix-docs/Software%20Guide/Bonsai.ONIX/index.html).

**Behavioral analysis.** For comparison between nonimplanted and implanted animals, markerless 3D tracking[17] from an array of five cameras (Extended Data Fig. 5) was used to measure head position and orientation in 3D (Fig. 2). For implanted animals where no direct comparison to nonimplanted mice was performed (Fig. 3), head position and 3D-tracking data from the headstage were aligned in time and resampled to 100 Hz for further analysis. In all cases, distributions for running speed, head posture and location were plotted from the resulting 3D position and pitch/yaw data. For comparisons between implanted and nonimplanted animals, occupancy was computed in a $20 \times 20$ grid spanning the entire maze, excluding periods where the mouse was stationary and occupancy distributions were compared, excluding one home position where each mouse spent a higher proportion of time. Maze occupancy was compared by computing their Shannon entropy and re-sampling in time bins of ~1 min to compute bootstrap samples. The entropy of the occupancies was statistically indistinguishable within 4 h of standard tether versus 4 h of ONIX tether data, with median entropies of 0.273 versus 0.287 bit (normalized against uniform distributions with equal number of bins[61]) and 95% confidence bounds of 0.208–0.369 versus 0.224–0.383 bit. The entropy of the spatial occupancy for the classic headstage epochs had a median of 4.217 bit with a confidence interval of 3.998–4.381 bit. Heading distributions (Fig. 2c) were compared to the same method, but in $40 \times 40$ bins spanning 0–360° yaw and ±45° pitch.

**Reporting summary**
Further information on research design is available in the Nature Portfolio Reporting Summary linked to this article.

## Data availability
Experimental data from Figs. 2 and 3 are available on figshare at https://doi.org/10.6084/m9.figshare.27242340.v1 (ref. 72) and https://doi.org/10.6084/m9.figshare.26391160 (ref. 73). Additional example data recorded with the system can be made available upon request.

## Code availability
The full ONI specification is available at https://github.com/open-ephys/ONI. Design documents for the described ONIX hardware are available as follows: host interface (https://github.com/open-ephys/onix-fmc-host); breakout board (https://github.com/open-ephys/onix-breakout); 64-channel Intan headstage (https://github.com/open-ephys/onix-headstage-64); and Neuropixels headstage (https://github.com/open-ephys/onix-headstage-neuropix1). Software, along with extensive hardware and API documentation, is available at https://open-ephys.github.io/onix-docs. Firmware can be made available upon request as configurable IP blocks. Unless specified in the respective repositories, all material is distributed under the creative commons CC BY-NC-SA 4.0 license and is therefore free to adapt and to share with appropriate attribution and under the same license for noncommercial purposes. Data plotting was performed in MATLAB (v.2023 and 2024) and Python (v.3.6.15) and no custom algorithms were used.

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

## Acknowledgements

We thank the Open Ephys Production Site team for beta-testing, E. J. Dennis, A. Blot for additional system testing, J. Alves da Silva and H. Stensola for testing the 3D tracking system and A. Bahle for help with testing the Neuropixels headstage. We thank A. Lee and E. J. Dennis for comments on the paper. We acknowledge the following funding sources: J.P.N.: NIH Posdoctoral Ruth L. Kirschstein National Research Service Award 1F32MH107086-01, the Center for Brains, Minds and Machines at MIT, funded by NSF STC award CCF-1231216, and NIH 1R44NS127725-01 to Open Ephys. J.Z.: NIH 1R21EY028381, NIH R01MH118928. T.H.: Picower Fellowship by JPB Foundation and MIT Picower Institute, Brain Science Foundation Research Grant Award, Kavli-Grass-MBL Fellowship by Kavli Foundation, Grass Foundation, and Marine Biological Laboratory, Osamu Hayaishi Memorial Scholarship for Study Abroad, Uehara Memorial Foundation Overseas Fellowship and Japan Society for the Promotion of Science Overseas Fellowship. M.-S.H.v.d.G.: Mathworks Graduate Fellowship. M.T.H.: NIH R01NS106031 and R21NS103098. M.A.W.: National Science Foundation STC award CCF-1231216 and NIH TR01-GM10498, NIH R01MH118928 and Picower Institute Innovation Fund. J.V.: NIH 1K99NS118112-01 and Simons Center for the Social Brain at MIT postdoctoral fellowship. This research was partially funded by the Howard Hughes Medical Institute at the Janelia Research Campus. A.L. and J.H.S. thank the Allen Institute founder P. G. Allen for his vision, encouragement and support.

## Author contributions

ONI concept: J.P.N.; ONI spec and API: A.C.-L., J.P.N. and G.L.; firmware prototype: J.Z. and J.P.N.; firmware implementation: J.P.N., J.Z. and A.C.-L.; headstage hardware development: J.P.N., J.Z. and A.C.-L.; headstage beta-testing: M.-S.H.v.d.G., N.J.M. and T.H.; commutator: J.V. and J.P.N.; mouse electrophysiology: J.P.N., J.V. and N.J.M.; Neuropixels experiments: J.H.S. and A.L.; behavior comparison experiments: J.P.N, J.V. and N.J.M.; data analysis: J.P.N., J.V. and J.S.; Miniscope recordings: T.H., J.Z. and J.V.; documentation: J.P.N., A.C.-L. and A.H.L.; beta-testing coordination: J.P.N and A.H.L.; and paper: J.V., J.P.N., M.T.H. and M.A.W., with input from all authors.

## Competing interests

J.P.N. is the president and J.V. and J.H.S. are board members of Open Ephys, a public benefit workers cooperative in Atlanta GA. F.C. is the

founder of the Open Ephys Production Site in Lisbon Portugal. A.C.L. and F.C. are, and A.H.L was employed at the Open Ephys Production Site in Lisbon Portugal. G.L. is a director of NeuroGEARS. The other authors declare no competing interests.

## Additional information

**Extended data** is available for this paper at https://doi.org/10.1038/s41592-024-02521-1.

**Correspondence and requests for materials** should be addressed to Jakob Voigts.

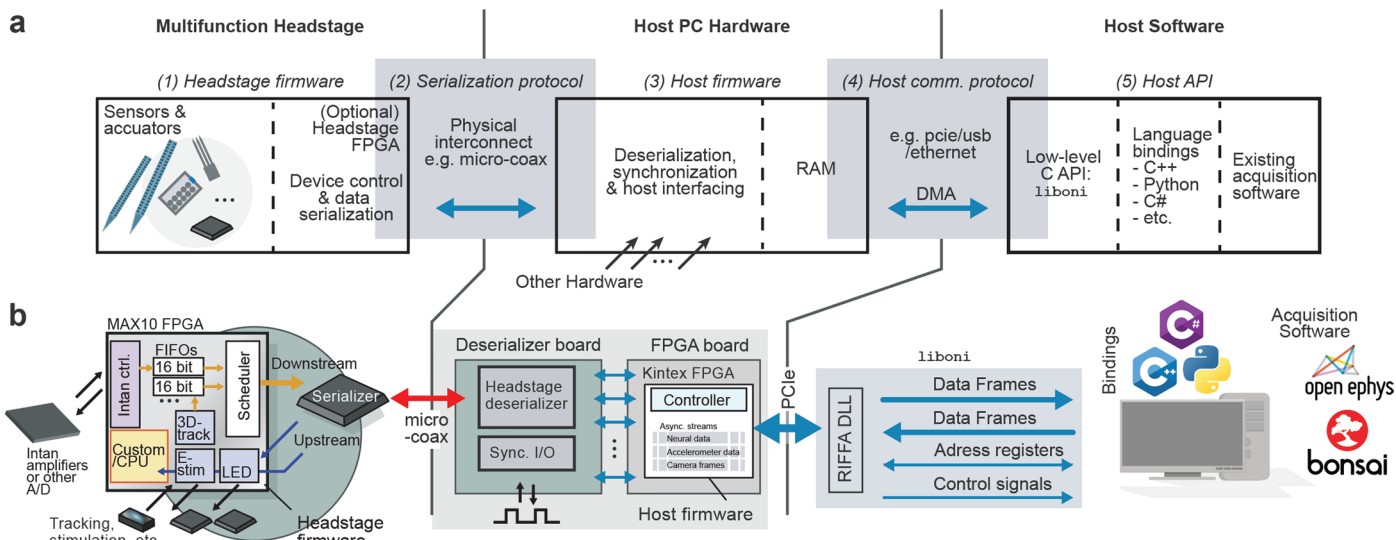

**Extended Data Fig. 1 | ONIX system block diagram. a**, The ONIX system architecture consists of 5 building blocks: (1) headstage firmware, (2) data serialization protocol, (3) host firmware, (4) host communication protocol, and (5) application programming interface (API). The firmware modules and host API (open black boxes) are both specified and implemented within this project. These are reusable across host hardware and physical interfaces (for example USB, ethernet, PCIe, etc). The serialization and host communication protocols (gray boxes) are generically specified by ONI. This project focuses on a micro-coaxial serializer to implement this spec. Other communication options would require custom, ONI-conforming firmware to be written. **b**, Each element of the architecture in panel (**a**) applied to the 64-channel Intan headstage used in Figs. 1–3. The serialization protocol and host communication protocol are implemented using a micro-coaxial serialization link and PCIe, respectively.

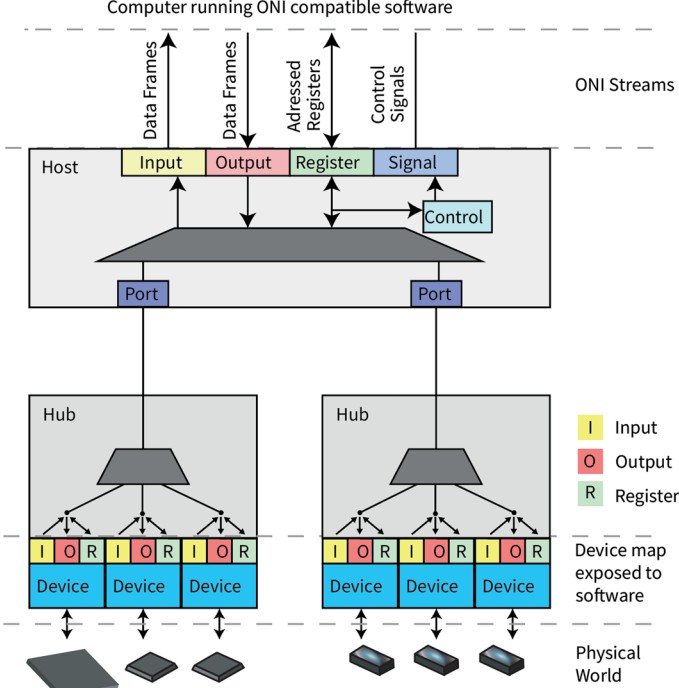

**Extended Data Fig. 2 | ONI communication block diagram.** Hubs (for example headstages, miniscopes, etc.) aggregate data and control sensors and actuators that interact with the world. Each sensor is provided with a separate, low-latency FIFO and control block. Configuration commands are demultiplexed from the host communication link to control each device on the headstage. Sensor data is multiplexed into a shared high-bandwidth link by a multiplexer on each hub. Low-bandwidth bidirectional configuration data is also transmitted over this link. In the hardware described in this paper, this link is implemented over a micro-coaxial serialization link, but could also be implemented using other wired or wireless links. Sensor data is demultiplexed by the host serial interface to create a local representation of the hub. This allows transparent control of each hub from the host computer, and low-level communication with the physical headstage via the serialized link is hidden by the firmware. Sensor and configuration data are streamed to and from the host PC using a high-speed bus, such as PCIe or USB3.0. The same API is used regardless of the physical communication interface. The appropriate drivers and translation layers are dynamically loaded for each physical interface that is used.

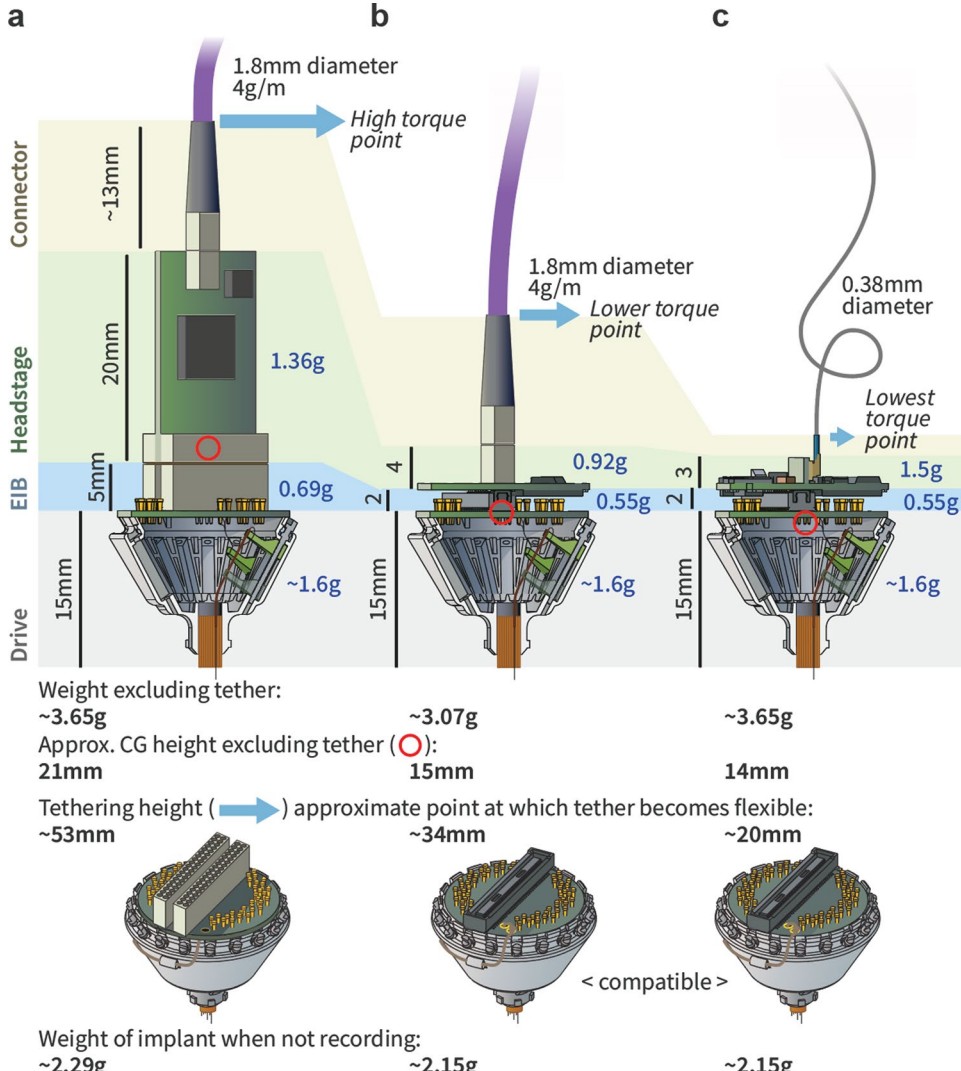

**Extended Data Fig. 3 | Relative weights and torque arms of comparable headstages.** Comparisons are between headstages on the same low-profile 16 tetrode drive implant[1]. **a**, Current Intan-based headstage or comparable system. Omnetics connectors, vertical headstage layout, and tether all contribute to height and long moment arm, so that sideways forces on the tether exert strong rotational forces on the animal's head. **b**, Best-possible low-profile headstage with current digital tethers, assuming a single, flat PCB design equivalent to the ONIX headstages described here. The digital tether still leads to a significant moment arm. **c**, In addition to a low-profile headstage, the ONIX micro-coax further lowers the torque arm, and also practically eliminates torque loads by virtue of an extremely light and flexible tether.

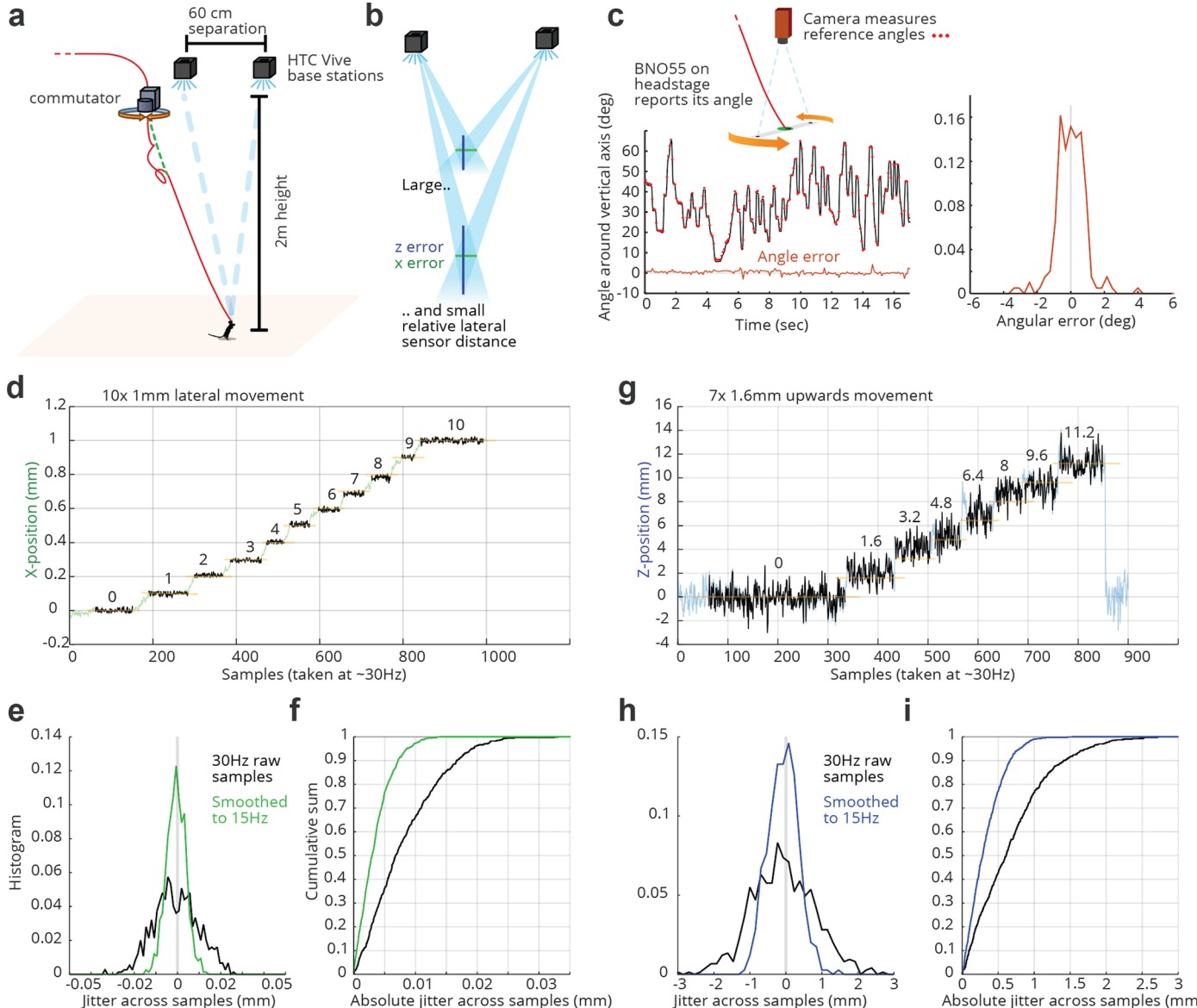

**Extended Data Fig. 4 | Precision of the 3D-tracking and 3D-pose measurements. a**, Experimental setup: A ONIX headstage was moved by known amounts at a 2-meter distance from the base stations. The base station pair was separated by 60 cm. **b**, Coordinates are derived from angular tracking data relative to the known base station positions. In our case (60 cm separation at a 2 m distance), this results in worse resolution in the height/Z direction than in the lateral/XY direction. **c**, Practically achievable absolute angular error of the BNO055 sensor on the headstages in dynamic environments without explicit calibration procedure. The absolute angle of a headstage was measured with an overhead camera, and the headstage was rotated randomly by hand in a non-optimized lab environment without any magnetic shielding. The ~2 degrees error

range is consistent with the reported accuracy for the BNO055. **d**, Raw position measurements for known step displacements. **e**, Measurement jitter relative to known positions. In addition to the raw sample data, smoothed data at 15 Hz are plotted. **f**, Resulting cumulative distribution of absolute jitter. **g-i**, same as c-e but for height/Z measurements. The setup used here with large distance from the base stations and small distance between the base stations represents a worst-case scenario and demonstrates degraded Z resolution. Higher Z resolution can be achieved at the same distances by increasing the spacing between the base stations above 60 cm or decreasing the height of the base stations relative to the headstage.

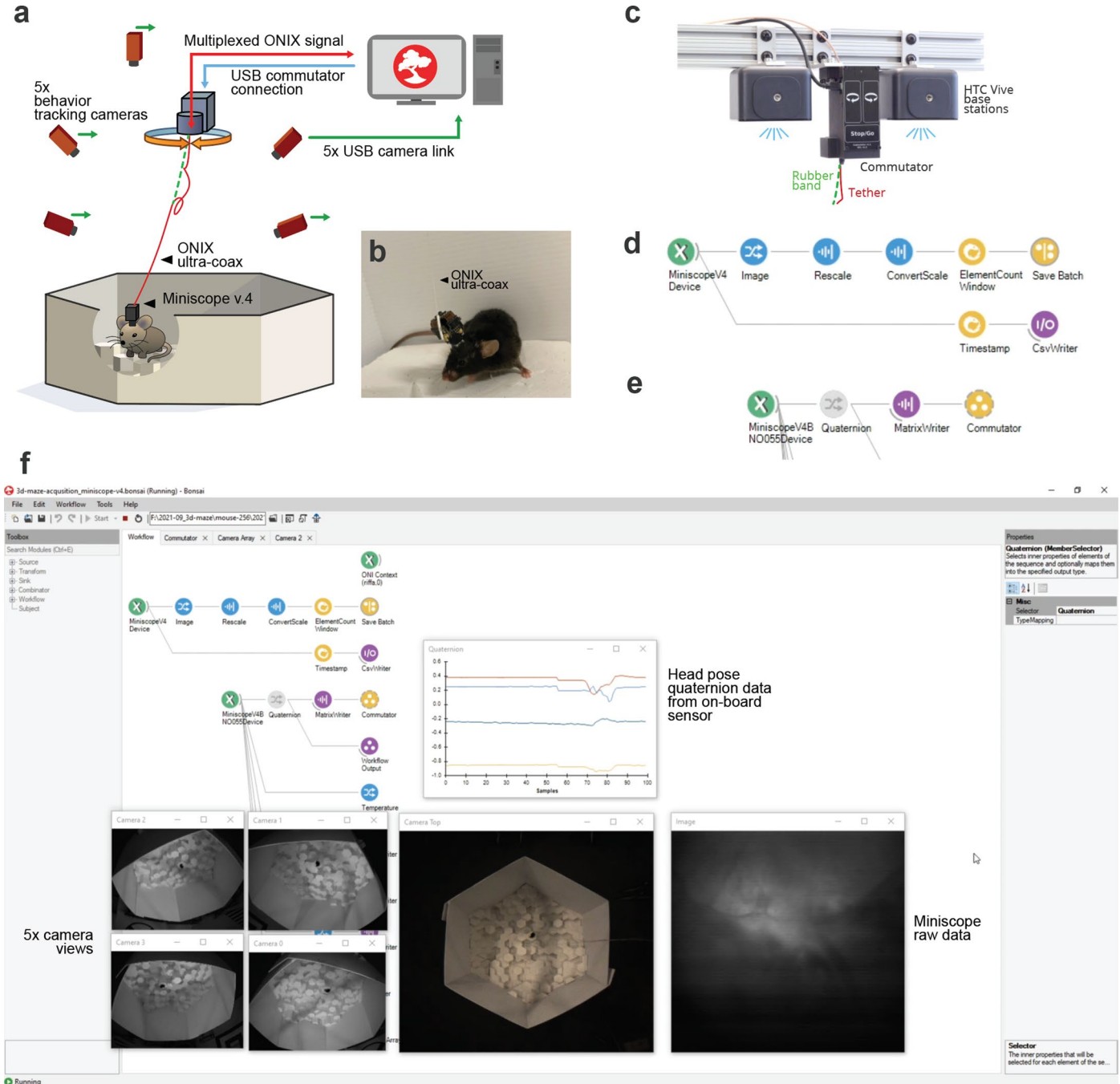

**Extended Data Fig. 5 | Example setup and Bonsai workflow for simultaneous neural recording and behavior tracking. a**, A mouse with a Miniscope implant explores the same arena described in Figs. 2 and 3, and is being tracked by 4 side-mounted, and one overhead camera. The experiment is performed using the Bonsai[2] software. In addition to the headstage connection (red), Bonsai controls the motorized commutator (blue) via a serial-over-USB link, as well as the 5 cameras (green) using the camera vendor API over USB. **b**, Mouse with a Miniscope v.4 implant[3], controlled by Bonsai via the ONIX system. **c**, The motorized commutator and 3D-tracking base stations. **d**, Excerpt of the Bonsai workflow: Data arrives via the 'MiniscopeV4' node (green), which communicates with a standard Miniscope through the ONIX micro-coax, is rescaled using an image processing nodes (blue) and saved to disk (yellow). A separate data

path assigns time stamps to frames, synchronization with behavior cameras or other data. **e**, Bonsai workflow used to drive the commutator (Fig. 1). The head orientation quaternion output of the 'MiniscopeV4 BNO55' node (green) is one of the outputs of the BNO55 chip on the headstage (shared across the Miniscope, and the 64-channel Intan, as well as Neuropixel headstages). The quaternion is the saved to disk as raw data (purple), and sent to the 'Commutator' node (yellow), which drives the commutator to follow the animal's rotation and remove any twisting of the tether. **f**, Example screenshot from Bonsai showing simultaneous data acquisition from all sources. Example workflows are available on the ONIX GitHub repository (https://open-ephys.github.io/onix-docs/Software%20Guide/Bonsai.ONIX/Bonsai%20Examples/index.html).

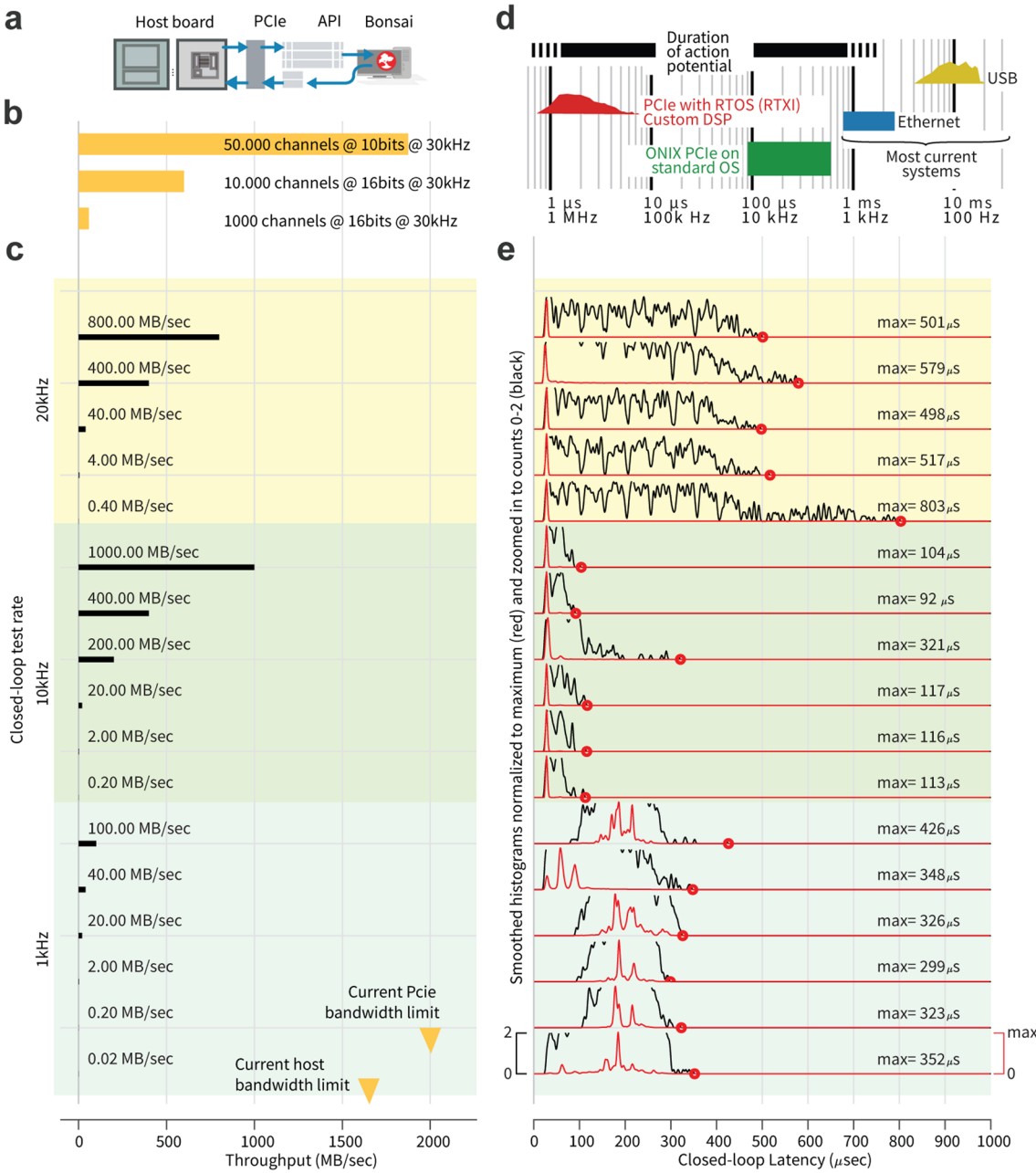

**Extended Data Fig. 6 | Host interface latency. a**, Latencies were measured for a closed-loop round-trip from the PCIe host to C software and back. **b**, Example theoretical data rate requirements for some high channel count recordings at 16 or 10 bit sample depth. Actual data rate requirements may differ depending on communication overhead. Future work could reduce the bandwidth requirements by compressing data on the headstage[4,5]. **c**, Different frequencies at which the closed-loop tests were run, each with increasing buffer sizes and resulting theoretical bandwidth. Note that the actual bandwidths can be limited by the headstage interconnect. **d**, Overview of latency ranges achievable with other technologies. **e**, Distribution of measured round-trip latency for all settings on a Windows 10 desktop computer with an Intel i5 CPU. The lowest latencies were measured with a 10 kHz rate. Results will differ on different hardware/OS, and depend on system load.

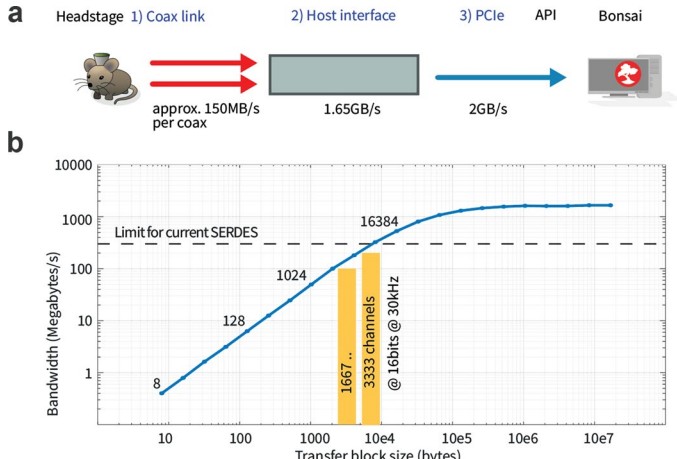

**Extended Data Fig. 7 | Real-world data rates of the ONIX system. a**, Overview of the system bandwidths. Bandwidth limits apply to different parts of the system: 1) The current serial over coax (Serializer-deserializer, SERDES) chip used on the headstages described here (Texas Instruments FPD-Link III for r 1-MP/60-fps), using a 100 MHz clock rate on a 12-bit interface, results in 1200Mbps (150MBps) bandwidth. The currently highest used bandwidth is ~48MBps on the Neuropixels headstage. Future implementation with other SERDES chips can improve this bandwidth, with the current headstage deserializers imposing a limit of ~300MBps. 2) The host interface currently runs an internal 64 bit bus at 250 MHz and is capable of 1.65GB/s. 3) Finally, the 4-lane PCIe interface used by ONIX is theoretically capable of ~2GB/s, of which we can currently use 1.65GB/s due to protocol overhead. It would be trivial to expand to higher lane counts in the future. **b**, To measure real-world throughput, the ONIX host has a load-testing FPGA-core with a configurable packet size and rate, which result in a given bandwidth. For a set of given transfer block sizes (here, powers of 2 starting at 64 bytes were used), we gradually increased the load bandwidth until the output FIFO, which is implemented on external RAM, started to fill, indicating that the transfer speed couldn't cope with the data production. The resulting curve therefore corresponds to empirically achievable data rates. Measured bandwidth from data generated in a test device as a function of transfer block size. Yellow: Approximate data rate requirement for example numbers of electrophysiology channels, sampled at 30KHz and 16 bit with no optimization.

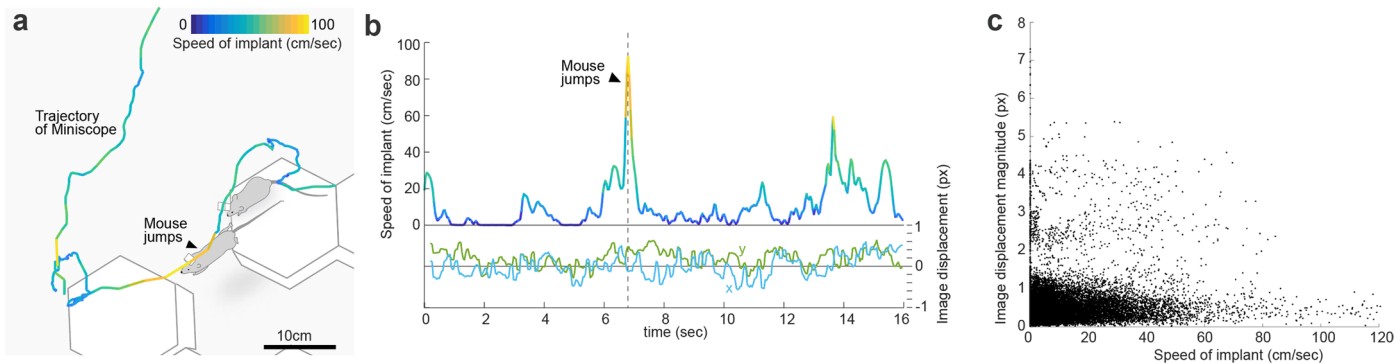

**Extended Data Fig. 8 | High-speed, high-acceleration behavior like jumping does not lead to large brain motion. a**, Excerpt from an experiment in which a mouse explored a 3D arena as described in the main experiment, with a Miniscope implant in dorsal Hippocampus. The 16 second excerpt shows the mouse jumping from one pillar onto a neighboring one at a distance of ~8.5 cm.

We performed markerless tracking of the implant[6], and measured brain motion by computing the displacement of the Miniscope images using an fft-based image stabilization algorithm[7]. **b**, Speed of the implant and brain motion for the excerpt. **c**, Speed of the implant shows no positive correlation with brain displacement.

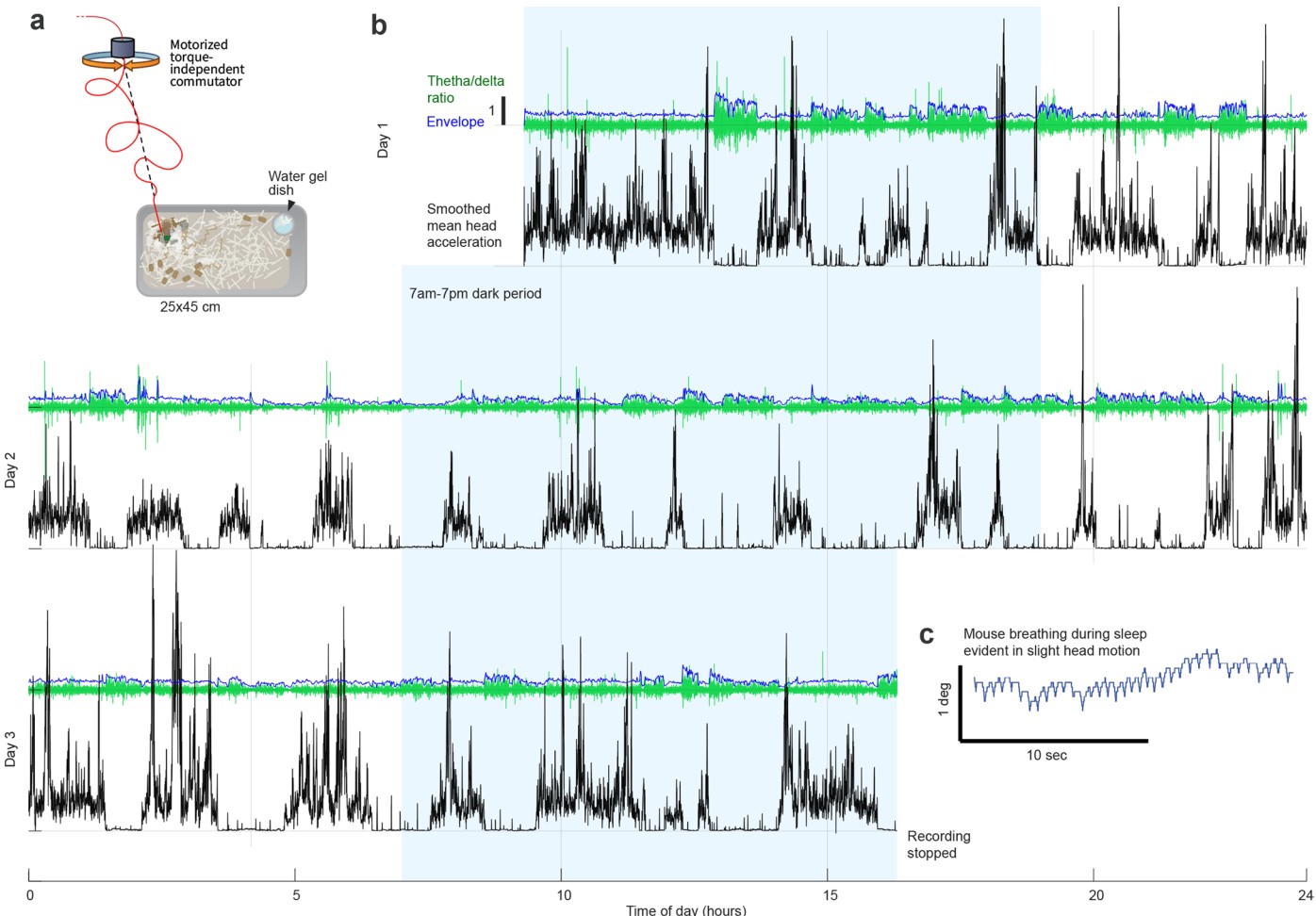

**Extended Data Fig. 9 | Stable long-term electrophysiology without experimenter intervention. a**, A mouse was placed in a large home cage and electrophysiology from a laminar probe in prefrontal cortex was recorded for ~55 hours in a mouse holding room with an automated 12-hour light cycle without experimenter intervention. We observed no tangling or twisting of the recording tether. **b**, Neural data was post-processed in the same way as for the other experiments, and data from one channel was bandpass filtered in the 0.1-2, 6–10 and 30–50 Hz bands to compute spectral power across awake and sleep phases. Behavioral activity levels were quantified by smoothing the norm of the 3-dimensional acceleration vector from the headstage at 1 Hz. **c**, during sleep periods, a clear breathing rhythm could be observed in the IMU head angle data recorded on the headstage.

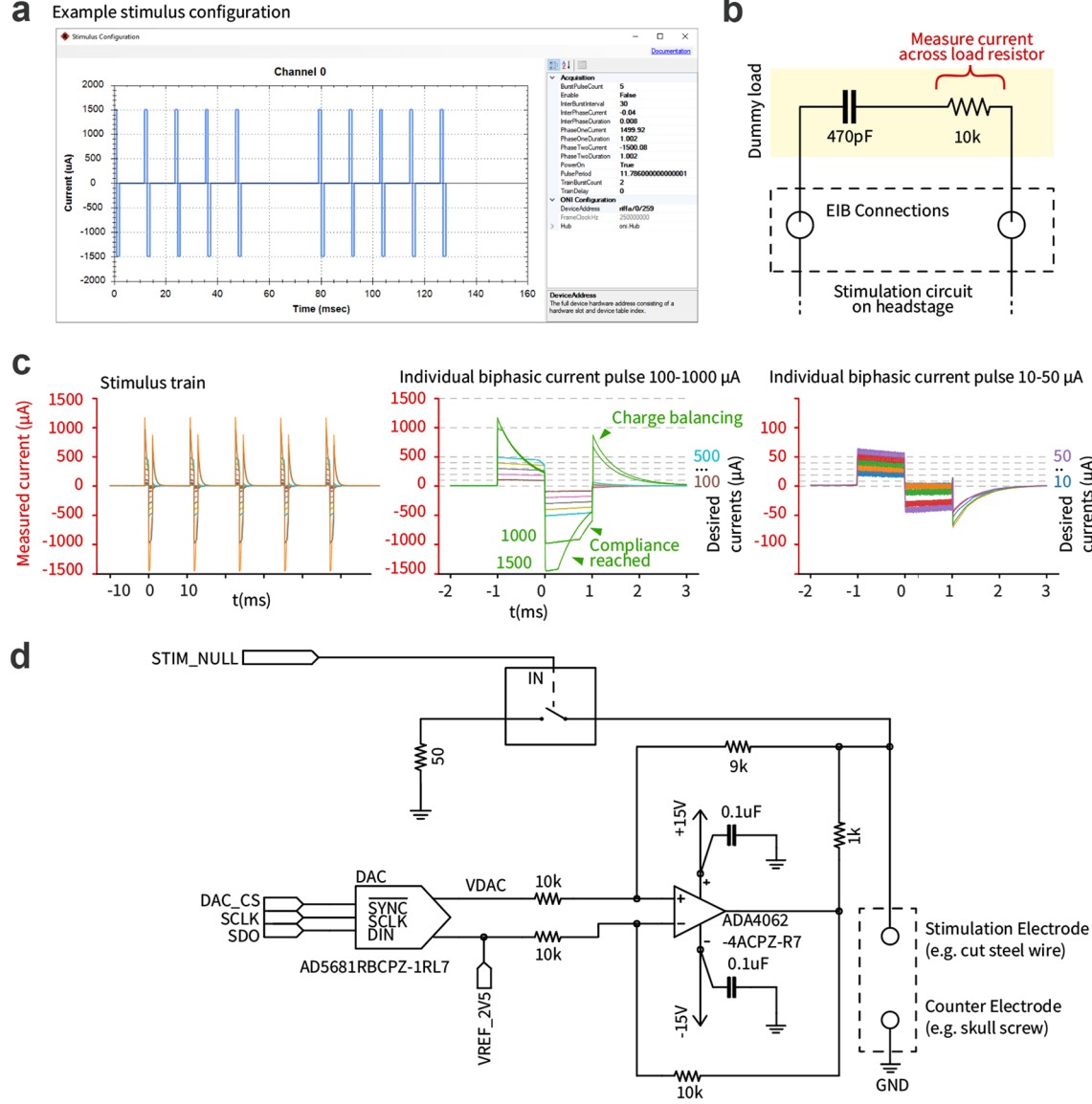

**Extended Data Fig. 10 | Integrated electrical microstimulation circuit on ONIX's 64-channel Intan headstage. a**, example pulse train configuration via graphical user interface in Bonsai. **b**, Setup for characterizing the circuit: The stimulator was used to drive biphasic current pulses into a model load (47 nF capacitor in series with 10kOhm resistor to ground), representing a high impedance stimulation scenario. The voltage across the 10 kOhm resistor was measured with an oscilloscope to monitor the delivered current. Pulse trains consisted of 1 ms per phase, biphasic, positive first pulses with inter-pulse interval of 10 msec (83 Hz pulse to pulse period). **c**, Measurement results: the amplitude of both phases was varied from ±10 uA to ±1.5 mA. For pulse-based stimulation, each phase must satisfy the following inequality to be achievable with the system's maximum voltage: $|I_{stim} * (R_{electrode} + t_{phase}/C_{electrode})| <= 15$ V. In our test setup, for ±500 uA, this results in -15.6 V per phase which is why voltage

compliance was reached for this target current. If the compliance voltage is reached, or pulses are delivered with asymmetric positive and negative charges, a charge imbalance will occur, resulting in a residual electrode voltage following the conclusion of the stimulus. The large artifacts following such failed stimuli result from the onboard charge balancing circuit removing this voltage after each stimulation pulse concludes. The mild slopes on the successfully delivered pulses are due to the finite voltage slew rate of the stimulation circuit. The stimulus current is recorded though an auxiliary channel on the Intan chip for verification of stimulus current waveforms. **d**, Simplified electrical stimulation circuit. A DAC controls an improved Howland current pump. Electrodes are discharged in between pulses to reduce artifacts and impose charge balancing. Control signals provided by the headstage FPGA are shown to the left. Elements are referred to in the text using their labels.

# Reporting Summary

## Statistics

For all statistical analyses, confirm that the following items are present in the figure legend, table legend, main text, or Methods section.

| n/a | Confirmed | |
|---|---|---|
| ☐ | ☒ | The exact sample size (*n*) for each experimental group/condition, given as a discrete number and unit of measurement |
| ☐ | ☒ | A statement on whether measurements were taken from distinct samples or whether the same sample was measured repeatedly |
| ☐ | ☒ | The statistical test(s) used AND whether they are one- or two-sided  <br> *Only common tests should be described solely by name; describe more complex techniques in the Methods section.* |
| ☐ | ☒ | A description of all covariates tested |
| ☒ | ☐ | A description of any assumptions or corrections, such as tests of normality and adjustment for multiple comparisons |
| ☐ | ☒ | A full description of the statistical parameters including central tendency (e.g. means) or other basic estimates (e.g. regression coefficient) AND variation (e.g. standard deviation) or associated estimates of uncertainty (e.g. confidence intervals) |
| ☐ | ☒ | For null hypothesis testing, the test statistic (e.g. *F*, *t*, *r*) with confidence intervals, effect sizes, degrees of freedom and *P* value noted  <br> *Give P values as exact values whenever suitable.* |
| ☒ | ☐ | For Bayesian analysis, information on the choice of priors and Markov chain Monte Carlo settings |
| ☒ | ☐ | For hierarchical and complex designs, identification of the appropriate level for tests and full reporting of outcomes |
| ☒ | ☐ | Estimates of effect sizes (e.g. Cohen's *d*, Pearson's *r*), indicating how they were calculated |

*Our web collection on statistics for biologists contains articles on many of the points above.*

## Software and code

Policy information about availability of computer code

| Data collection | Data was collected using our own software (github repositories are referred to in the text), and software that we wrote as part of the Bonsai programming language (also available on github and referenced in the text), as described in the manuscript. All source code is available as stated in the text. |
|---|---|
| Data analysis | Data plotting was performed in Matlab, versions 2023 and 2024, and Python (ver. 3.6.15) no custom algorithms were used. |

For manuscripts utilizing custom algorithms or software that are central to the research but not yet described in published literature, software must be made available to editors and reviewers. We strongly encourage code deposition in a community repository (e.g. GitHub). See the Nature Portfolio guidelines for submitting code & software for further information.

## Data

Policy information about availability of data

All manuscripts must include a data availability statement. This statement should provide the following information, where applicable:
- Accession codes, unique identifiers, or web links for publicly available datasets
- A description of any restrictions on data availability
- For clinical datasets or third party data, please ensure that the statement adheres to our policy

Experimental data from Figure 3 is available on figshare at 10.6084/m9.figshare.26391160. Additional example data recorded with the system can be made available upon request.

## Human research participants

Policy information about studies involving human research participants and Sex and Gender in Research.

| | |
|---|---|
| Reporting on sex and gender | N/A |
| Population characteristics | N/A |
| Recruitment | N/A |
| Ethics oversight | N/A |

Note that full information on the approval of the study protocol must also be provided in the manuscript.

# Field-specific reporting

Please select the one below that is the best fit for your research. If you are not sure, read the appropriate sections before making your selection.

☒ Life sciences  ☐ Behavioural & social sciences  ☐ Ecological, evolutionary & environmental sciences

For a reference copy of the document with all sections, see nature.com/documents/nr-reporting-summary-flat.pdf

# Life sciences study design

All studies must disclose on these points even when the disclosure is negative.

| | |
|---|---|
| Sample size | We applied sample sized commonly used to assess whether a data acquisition system functions reliably. No power size calculations were used for this. |
| Data exclusions | No data exclusion was applied. |
| Replication | This study is based on performance metrics achievable with a new method. Other than the sample sizes used to ensure the robustness of these metrics, replication is no applicable in this context. |
| Randomization | Randomization is not applicable to a study establishing the basic function of a new method. |
| Blinding | There were no significant interactions between the users and the research animals that could be blinded. |

# Reporting for specific materials, systems and methods

We require information from authors about some types of materials, experimental systems and methods used in many studies. Here, indicate whether each material, system or method listed is relevant to your study. If you are not sure if a list item applies to your research, read the appropriate section before selecting a response.

### Materials & experimental systems

| n/a | Involved in the study |
|---|---|
| ☒ | ☐ Antibodies |
| ☒ | ☐ Eukaryotic cell lines |
| ☒ | ☐ Palaeontology and archaeology |
| ☐ | ☒ Animals and other organisms |
| ☒ | ☐ Clinical data |
| ☒ | ☐ Dual use research of concern |

### Methods

| n/a | Involved in the study |
|---|---|
| ☒ | ☐ ChIP-seq |
| ☒ | ☐ Flow cytometry |
| ☒ | ☐ MRI-based neuroimaging |

## Animals and other research organisms

Policy information about studies involving animals; ARRIVE guidelines recommended for reporting animal research, and Sex and Gender in Research

| | |
|---|---|
| Laboratory animals | ChAT-IRES-Cre transgenic, PWK/PhJ x C57bl6, and C57bl6, all Jackson labs. Mice were aged between 8 and 14 months, and both male and female mice were used. Animals were housed in accordance with institutional guidelines approved by the internal IACUC at the |

Allen institute, MIT and HHMI Janelia research campus, in compliance with the standards set forth by the Association for Assessment and Accreditation of Laboratory Animal Care. This includes maintaining temperatures of 65-75°F (~18-23°C) with 40-60% humidity.

Wild animals

No wild animals were used in this study.

Reporting on sex

No effects related to sex were considered in this study.

Field-collected samples

No field collected samples were used in this study.

Ethics oversight

Massachusetts Institute of Technology (MIT), Allen Institute, and Janelia Institutional Animal Care and Use Committees.

Note that full information on the approval of the study protocol must also be provided in the manuscript.

