## [Peer Review File · Nature Methods]

ONIX: A unified open-source platform for multimodal neural recording and perturbation during naturalistic behavior

Corresponding Author: Dr Jakob Voigts

Version 0:

Decision Letter:

29th Nov 2023

Dear Dr Voigts,

Thank you for your patience during the review process. We were waiting for input from another reviewer, who unfortunately has not delivered. Your Article, "A unified open-source platform for multimodal neural recording and perturbation during naturalistic behavior", has now been seen by two reviewers. As you will see from their comments below, although the reviewers find your work of considerable potential interest, they have raised a number of concerns. We are interested in the possibility of publishing your paper in Nature Methods, but would like to consider your response to these concerns before we reach a final decision on publication.

We therefore invite you to revise your manuscript to address these concerns. Importantly, please make sure that sufficient information is provided, so that others can reproduce the system.

Link Redacted

We hope to receive your revised paper within 2-3 months. If you cannot send it within this time, please let us know. In this event, we will still be happy to reconsider your paper at a later date so long as nothing similar has been accepted for publication at Nature Methods or published elsewhere.

OPEN SCIENCE REQUIREMENTS

REPORTING SUMMARY AND EDITORIAL POLICY CHECKLISTS

Please note that these forms are dynamic ‘smart pdfs’ and must therefore be downloaded and completed in Adobe Reader. We will then flatten them for ease of use by the reviewers. If you would like to reference the guidance text as you complete the template, please access these flattened versions at <http://www.nature.com/authors/policies/availability.html>.

DATA AVAILABILITY

All novel DNA and RNA sequencing data, protein sequences, genetic polymorphisms, linked genotype and phenotype data, gene expression data, macromolecular structures, and proteomics data must be deposited in a publicly accessible database, and accession codes and associated hyperlinks must be provided in the “Data Availability” section.

Please include a “Data availability” subsection in the Online Methods. This section should inform readers about the availability of the data used to support the conclusions of your study, including accession codes to public repositories, references to source data that may be published alongside the paper, unique identifiers such as URLs to data repository entries, or data set DOIs, and any other statement about data availability. At a minimum, you should include the following statement: “The data that support the findings of this study are available from the corresponding author upon request”, describing which data is available upon request and mentioning any restrictions on availability. If DOIs are provided, please include these in the Reference list (authors, title, publisher (repository name), identifier, year). For more guidance on how to write this section please see: <http://www.nature.com/authors/policies/data/data-availability-statements-data-citations.pdf>

CODE AVAILABILITY

Please include a “Code Availability” subsection in the Online Methods which details how your custom code is made available. Only in rare cases (where code is not central to the main conclusions of the paper) is the statement “available upon request” allowed (and reasons should be specified).

MATERIALS AVAILABILITY

As a condition of publication in Nature Methods, authors are required to make unique materials promptly available to others

without undue qualifications.

ORCID

Nature Methods is committed to improving transparency in authorship. As part of our efforts in this direction, we are now requesting that all authors identified as 'corresponding author' on published papers create and link their Open Researcher and Contributor Identifier (ORCID) with their account on the Manuscript Tracking System (MTS), prior to acceptance. This applies to primary research papers only. ORCID helps the scientific community achieve unambiguous attribution of all scholarly contributions. You can create and link your ORCID from the home page of the MTS by clicking on 'Modify my Springer Nature account'. For more information please visit <http://www.springernature.com/orcid>.

Best regards,
Nina

Nina Vogt, PhD
Senior Editor
Nature Methods

Reviewers' Comments:

Reviewer #2:

Remarks to the Author:

In this manuscript, the authors describe a new open-source data acquisition system for neural recordings in freely moving mice. This system is designed to provide power and high-rate data collection from head-mounted recording devices – ephys or miniscopes – with minimal impact on mouse behavior. The solution involves the combination of multiple improvements to several components of the data collection pipeline, including serial transfer of data and power along a thin coaxial cable, and a closed-loop commutator that unwinds the cable based on input from head angle changes.

Together, this system enables long timescale recording and dramatically alters the movement patterns of mouse behavior in complex three-dimensional environments. Particularly impressive are the videos demonstrating the mouse leaping and climbing across terrain while neural activity is recorded using a nearly invisible cable. The data is thoroughly analyzed, and improvements are clearly described. The system complements two widely used technologies in mice – neuropixel probes and head-mounted miniature microscopes. If the system is as easy to set up and use as the paper suggests, it could be widely adopted. Finally, the paper is detailed, well-written and easy to follow.

Here are my concerns:

- 1) It is not clear what experiments this system allows that could not be done before. Do the authors see this as a product that represents an incremental improvement over existing systems, or does it provide a new framework for experiments? If the latter, which experiments does the system allow? I think this is a critical question for a journal like Nature Methods.
- 2) Related to item 1, I do not see a clear improvement in behavior beyond increased exploration. The authors motivate the development of the system by discussing applications for the study of natural behavior, including social behavior; however, they do not demonstrate any change in social behavior using this system. How feasible would it be to have one or more animals implanted with this device in a social setting? It seems like multiple problems could arise, including tangling or an alteration in treatment against the implanted individual.
- 3) It would be useful to have a parts list. Perhaps I missed this? Supplementary methods table one is not sufficient. Is the idea that this is a product that will be purchased from open ephys, or is this something that labs will build themselves? Perhaps they will do both, but either way, it would be valuable to provide a detailed parts list and fabrication instructions.
- 4) I imagine that the freedom of movement that the cable allows could lead to increased brain motion artifacts during imaging. Do the authors observe increased brain motion or other artifacts using this system as compared with existing systems? Can these artifacts be identified or corrected using the onboard accelerometers?
- 5) What is the reliability of this system? Which parts break first and when?

Reviewer #3:

Remarks to the Author:

The authors present ONI, a hard- and software specification for transferring multiplexed sensor data from a freely moving animal to a computer and send closed-loop instructions back at low (<1ms) latency. They further present an implementation, ONIX, and demonstrate its capabilities with HTC Vive and IMU based position and orientation tracking (30Hz) combined with 4 tetrodes as well as with a UCLA Miniscope and with a Neuropixel headstage. According to the manuscript, a central component of the ONIX implementation is a 0.31 mm thin coaxial fiber combined with a commutator on the stationary receptor end that corrects for tether torsion and thus allows for several hours of uninterrupted, unsupervised recording. The authors convincingly show that their system works for the advertised applications and has significantly less influence on the behavior of a freely moving animal than a "standard tether" (Intan "SPI Cable", 12 wires, 1.8 mm diameter). Running speed compared to an untethered animal decreased by a factor of 2, compared to a factor of 24 with a "standard tether".

I have the following comments and questions:

1. The author put considerable emphasis on the 0.31 mm coaxial cable and the comparison with the "standard tether". The "standard tether" is quite thick and stiff and considerable impact on the animals behavior is expected. In contrast to the implications of the manuscript, coaxial cables of the order of magnitude of 0.3 mm are already being used in the field, e.g. in references 45 and 47. It seems that an emphasis on the commutator would better underline the novelties of the ONIX system.
2. If I read Suppl. fig. 8 right, the main text should not claim 0.02 mm resolution for "6-degrees-of-freedom head pose". The maximum error is in z direction and seems more in the range of 2mm.
3. An estimate of the orientation accuracy should be given. Many publications like <https://doi.org/10.3390/s20143824> and other AHRS evaluations suggest that the error from sensor fusion of IMUs is substantial in dynamic environments, and the BNO055 will likely not exceed established algorithms. Since localisation is not the topic of this manuscript, a realistic estimation would not impair the impact of ONIX, but it would help experimenters to know what they can expect from a setup like the one presented.
4. Why could twisted pairs used with Neuropixels not be equipped with zero-torque commutation?
5. Suppl. fig. 8: "f-g" legend should be "f-h", or better "f,g,h"?

An open tethered bidirectional transmission system that can be flexibly adapted to a variety of sensor platform is certainly a timely and welcome tool for the important and growing field of behavioral quantification of freely moving animal. However a different focus on the novel elements might be advised.

Version 1:

Decision Letter:

Our ref: NMETH-A53635A

19th Jun 2024

Dear Dr. Voigts,

Thank you for submitting your revised manuscript "A unified open-source platform for multimodal neural recording and perturbation during naturalistic behavior" (NMETH-A53635A). It has now been seen by the original referees and their comments are below. The reviewers find that the paper has improved in revision, and therefore we'll be happy in principle to publish it in Nature Methods, pending minor revisions to satisfy the referees' final requests and to comply with our editorial and formatting guidelines.

TRANSPARENT PEER REVIEW

Nature Methods offers a transparent peer review option for new original research manuscripts submitted from 17th February 2021. We encourage increased transparency in peer review by publishing the reviewer comments, author rebuttal letters and editorial decision letters if the authors agree. Such peer review material is made available as a supplementary peer review file.

Please state in the cover letter 'I wish to participate in transparent peer review' if you want to opt in, or 'I do not wish to participate in transparent peer review' if you don't. Failure to state your preference will result in delays in accepting your manuscript for publication.

Please note: we allow redactions to authors' rebuttal and reviewer comments in the interest of confidentiality. If you are concerned about the release of confidential data, please let us know specifically what information you would like to have removed. Please note that we cannot incorporate redactions for any other reasons. Reviewer names will be published in the peer review files if the reviewer signed the comments to authors, or if reviewers explicitly agree to release their name. For more information, please refer to our <https://www.nature.com/documents/nr-transparent-peer-review.pdf> target="new">FAQ page.

ORCID

Best regards,
Nina

Nina Vogt, PhD
Senior Editor
Nature Methods

Reviewer #2 (Remarks to the Author):

The new additions to the revision are clear and impressive. The authors have addressed all my concerns. This is a nice paper.

Benjamin Scott
Boston University

Reviewer #3 (Remarks to the Author):

In the revised manuscript, the authors have answered all of my concerns. I have no further comments. This is a nice study.

Version 2:

Decision Letter:

17th Oct 2024

Dear Jakob,

I am pleased to inform you that your Article, "ONIX: A unified open-source platform for multimodal neural recording and perturbation during naturalistic behavior", has now been accepted for publication in Nature Methods. The received and accepted dates will be August 30th, 2023 and October 17th, 2024. This note is intended to let you know what to expect from us over the next month or so, and to let you know where to address any further questions.

Over the next few weeks, your paper will be copyedited to ensure that it conforms to Nature Methods style. Once your paper is typeset, you will receive an email with a link to choose the appropriate publishing options for your paper and our Author Services team will be in touch regarding any additional information that may be required. It is extremely important that you let us know now whether you will be difficult to contact over the next month. If this is the case, we ask that you send us the contact information (email, phone and fax) of someone who will be able to check the proofs and deal with any last-minute problems.

Please note that *Nature Methods* is a Transformative Journal (TJ). Authors may publish their research with us through the traditional subscription access route or make their paper immediately open access through payment of an article-processing charge (APC). Authors will not be required to make a final decision about access to their article until it has been accepted. [Find out more about Transformative Journals](https://www.springernature.com/gp/open-research/transformative-journals)

Best regards,
Nina

Nina Vogt, PhD
Senior Editor
Nature Methods

** Visit the Springer Nature Editorial and Publishing website at http://editorial-jobs.springernature.com?utm_source=ejP_NMeth_email&utm_medium=ejP_NMeth_email&utm_campaign=ejp_Nmeth for more information about our career opportunities. If you have any questions please click [here](mailto:editorial.publishing.jobs@springernature.com).**

permits use, sharing, adaptation, distribution and reproduction in any medium or format, as long as you give appropriate credit to the original author(s) and the source, provide a link to the Creative Commons license, and indicate if changes were made. In cases where reviewers are anonymous, credit should be given to 'Anonymous Referee' and the source.

Reviewers' Comments:

Reviewer #2:

Remarks to the Author:

In this manuscript, the authors describe a new open-source data acquisition system for neural recordings in freely moving mice. This system is designed to provide power and high-rate data collection from head-mounted recording devices – ephys or miniscopes – with minimal impact on mouse behavior. The solution involves the combination of multiple improvements to several components of the data collection pipeline, including serial transfer of data and power along a thin coaxial cable, and a closed-loop commutator that unwinds the cable based on input from head angle changes.

Together, this system enables long timescale recording and dramatically alters the movement patterns of mouse behavior in complex three-dimensional environments. Particularly impressive are the videos demonstrating the mouse leaping and climbing across terrain while neural activity is recorded using a nearly invisible cable. The data is thoroughly analyzed, and improvements are clearly described. The system complements two widely used technologies in mice – neuropixel probes and head-mounted miniature microscopes. If the system is as easy to set up and use as the paper suggests, it could be widely adopted. Finally, the paper is detailed, well-written and easy to follow.

We thank the reviewer for the constructive feedback, our revisions should address the remaining questions and concerns.

Here are my concerns:

1) It is not clear what experiments this system allows that could not be done before. Do the authors see this as a product that represents an incremental improvement over existing systems, or does it provide a new framework for experiments? If the latter, which experiments does the system allow? I think this is a critical question for a journal like Nature Methods.

Our system allows a completely new type of experiment: Large-scale neural measurements in small animals like mice without behavioral confounds over long times, and in large environments. Some aspects of these methods, e.g. long-term recordings with heavy stiff tethers in rats under near-constant supervision in small boxes (Dhawale et al. 2017 eLife), or recordings in large arenas in rats with battery-powered loggers (Bagi, Brecht & Sanguinetti-Scheck 2022 CurrBio), have been described previously. However, the combination of these different component methods to fully enable what the field wants/needs (long duration recordings from mice in large arenas with naturalistic behavioral expression) have simply not been possible. Crucially, for mice there is no current method that can record at all during unencumbered freely moving behaviors like we demonstrate here.

Additionally, the closed-loop latencies of our system (reacting to spikes with sub-millisecond round-trip delays) were previously only possible when programming the closed-loop algorithms in specialized DSP hardware. Our system allows the use of regular programming languages like C, Julia, or even python (Fig.1e, Suppl. Fig. 6). We now make this categorical difference clearer in the text in the section on closed-loop latencies: “This level of latency, otherwise only achievable on specialized operating systems⁵² or hardware⁵³, enables scientists to develop high-performance yet replicable closed-loop systems.”

In addition to the RSC recording where we show spontaneous jumping during a ~7.5 hour session, we now include a new experiment where we left a mouse plugged in for ~55 hours without any need for experimenter intervention, to further showcase another type of experiment that was impossible prior to our new technology: Only the combination of a very thin tether with an error-free real-time tracking of

mouse rotation could keep the recording completely tangle free while a mouse behaved freely for this long.

These experiments show that completely new types of studies are now possible, bringing electrophysiology or imaging capabilities to any behavior that unfolds over spaces bigger than a cage, over timescales longer than a few hours, and/or requires unencumbered behavior.

2) Related to item 1, I do not see a clear improvement in behavior beyond increased exploration.

Spontaneous exploration over long timescales is not just the aspect of behavior most impacted by current neuroscience methods, but is the central aspect of awake mouse behavior. In the wild, mice explore and forage in large environments and establish large territories. A majority of essential behaviors, from predator avoidance, to foraging, to mate selection and other conspecific interactions are expressed through locomotion in these environments (P Crowcroft, ‘Spatial Distribution of Feeding Activity in the Wild House-Mouse (*Mus Musculus L.*)’, 1959 and Territoriality in Wild House Mice, *Mus musculus L.*, 1955), and moving their body through space is the basis for most, if not all, other behavior.

The importance of studying exploratory behavior is evident from the many studies that use non-headfixed animals to gain insight into brain function. However, due to lack of adequate methods, this is currently done in small unnatural boxes, with limited mobility, or without any electrophysiology. To just list a few examples of studies from 2023 that employ spontaneous behavior but had to compromise on arena size, behavior duration, and/or omitted measurements of brain activity:

- Gschwind et al.: Hidden behavioral fingerprints in epilepsy,
- Markowitz et al.: Spontaneous behaviour is structured by reinforcement without explicit reward,
- Shamash et al.: Mice identify subgoal locations through an action-driven mapping process,
- Chen et al.: Rearing behaviour in the mouse behavioural pattern monitor distinguishes the effects of psychedelics from those of lisuride and TBG,
- Dasgupta et al.: Wireless monitoring of respiration with EEG reveals relationships between respiration, behaviour and brain activity in freely moving mice,
- We also cited a few of what we felt were the most interesting examples of spatial exploration behavior in the main text, e.g. play (Reinhold et al. 2019).

Our technology now makes it possible for the first time to perform such studies with high-throughput brain measurements, and without limiting the behavior itself. This is a major advance and will have wide impact in neuroscience. We have made this advance compared to existing methods more explicit in the revised discussion section.

The authors motivate the development of the system by discussing applications for the study of natural behavior, including social behavior; however, they do not demonstrate any change in social behavior using this system. How feasible would it be to have one or more animals implanted with this device in a social setting? It seems like multiple problems could arise, including tangling or an alteration in treatment against the implanted individual.

Regarding social behavior, we are not currently running any studies that require recordings in a social context, but the data already present in the paper, including a new demonstration of 55 hours of uninterrupted recording (also included here), show that such experiments are in principle possible and will benefit from the new technology described here, e.g. the ability to record during unencumbered behavior and for days rather than a few hours. The compatibility of tethers or fibers with social behavior has been demonstrated by multiple studies, e.g. Marlin et al 2015 that we cite, and recording from an animal in such a setting would not differ from the recordings we show. Multiple recorded animals would indeed be subject to twisting of the tethers, and we now explicitly mention this limitation.

New supplementary Figure 10, showing an uninterrupted 55 hour recording.

3) It would be useful to have a parts list. Perhaps I missed this? Supplementary methods table one is not sufficient. Is the idea that this is a product that will be purchased from open ephys, or is this something that labs will build themselves? Perhaps they will do both, but either way, it would be valuable to provide a detailed parts list and fabrication instructions.

All design files, including parts lists (BOM – Bill of materials) are indeed linked in the ‘Code and design file availability’ section, but this was not immediately obvious from the main text. We have now added a clear reference to this section in the main text: ‘Parts lists and design documents for the system are available via <https://github.com/open-ephys>, see the ‘Code, data, and design file availability’ section for detail.’ For example, the design files and parts for the 64 channel headstage are found at <https://github.com/open-ephys/onix-headstage-64/tree/main/headstage-64>. This information is sufficient to build the system. Firmware will be made available for non-commercial use and all software is open-source. The main use of the design files is to make it possible for labs to design variants of headstages, possibly with the help of engineering core facilities or in collaboration with engineering experts. While the hardware could be built by individual labs, the complexity of the designs makes this advisable only for technically skilled labs, and this would likely cost more than getting the hardware through Open Ephys or other manufacturers.

4) I imagine that the freedom of movement that the cable allows could lead to increased brain motion artifacts during imaging. Do the authors observe increased brain motion or other artifacts using this

system as compared with existing systems? Can these artifacts be identified or corrected using the onboard accelerometers?

This is an interesting suggestion; we have now tested this hypothesis in a new supplementary figure 9. There are two main sources of fast brain motion in imaging experiments:

1) Muscle strain of the mouse against the head-post or implant – for example in head-fixed mice, or if mice push the implant against walls. This is why brain motion in fully head-fixed mice is higher than in mice that can rotate their head freely (Voigts & Harnett 2019, Suppl. Fig.3). This effect would not be affected by our system, if anything the lower weight of the implant and reduced torque on the neck would decrease brain motion due to muscle strain.

2) Acceleration of the skull due to motion. The freedom of behavior with our system could in fact lead to stronger artifacts here. To quantify this effect, we performed a recording of a mouse exploring a 3D arena as described in the main experiment, with a Miniscope implant in dorsal Hippocampus that allowed us to directly observe brain motion:

Even during high-velocity jumps, we observed no significant increase in brain motion, and over the entirety of the recording, there was no correlation between brain displacement and movement speed. We have added this analysis as a new Supplementary Figure 9 and added a brief discussion of this observation to the main text: ‘We did not observe increased brain motion during high-speed motion or jumps (Suppl. Fig. 9)’.

5) What is the reliability of this system? Which parts break first and when?

This is a very relevant question, particularly since our system is making it possible to run experiments of unprecedented length. To summarize, in our experience, the only part of the system that can be expected to ‘break’ and needs to be replaced occasionally is the tether. In some experiments, for example the 3D arena, it is possible for mice to bite the tether if it is not properly counterbalanced due to user error. Luckily, being a 2-conductor coax cable this is a fairly cheap (under 50 USD in parts) replacement part. We have now also added a brief explanation of this to the main text.

Other than this, we have been using prototype versions of the system in multiple complete studies since 2019 with no spontaneous hardware failures. All connectors we use in the system are high cycle rated industry standard parts, and we designed the headstages so that the part of the connector that experiences wear is on the animal, and the part that does not experience wear is on the headstages. We have also tested the software stack for reliability by running very long acquisitions, including an experiment included here where a mouse stayed plugged in and was recorded from for ~55 hours straight (Suppl. Fig.10). We have now added a new section on system reliability to the supplement.

Reviewer #3:

Remarks to the Author:

The authors present ONI, a hard- and software specification for transferring multiplexed sensor data from a freely moving animal to a computer and send closed-loop instructions back at low (<1ms) latency. They further present an implementation, ONIX, and demonstrate its capabilities with HTC Vive and IMU based position and orientation tracking (30Hz) combined with 4 tetrodes as well as with a UCLA Miniscope and with a Neuropixel headstage. According to the manuscript, a central component of the ONIX implementation is a 0.31 mm thin coaxial fiber combined with a commutator on the stationary receptor end that corrects for tether torsion and thus allows for several hours of uninterrupted, unsupervised recording.

The authors convincingly show that their system works for the advertised applications and has significantly less influence on the behavior of a freely moving animal than a “standard tether” (Intan “SPI Cable”, 12 wires, 1.8 mm diameter). Running speed compared to an untethered animal decreased by a factor of 2, compared to a factor of 24 with a “standard tether”.

We thank the reviewer for their constructive feedback. Our revisions directly address all remaining questions and concerns.

I have the following comments and questions:

1. The author put considerable emphasis on the 0.31 mm coaxial cable and the comparison with the “standard tether”. The “standard tether” is quite thick and stiff and considerable impact on the animals behavior is expected. In contrast to the implications of the manuscript, coaxial cables of the order of magnitude of 0.3 mm are already being used in the field, e.g. in references 45 and 47. It seems that an emphasis on the commutator would better underline the novelties of the ONIX system.

The reviewer correctly points out that the UCLA miniscope system already uses coaxial cables, typically with an OD of 1.32 mm. These would indeed perform better than the SPI cables we used as a reference. However, these cables also lack commutation, similarly to the new Neuropixels tethers. Prior to our system these coax tethers were not available for use with electrophysiology or other high-bandwidth sensors. We have now added a specific mention of the Miniscope coax tethers to the section where we discuss the Neuropixels tethers.

We agree that a stronger emphasis of the commutator technology, and its synergy with thin tethers, better highlights the major technological novelty, and we have now added more specific descriptions of this in earlier in the introduction, added the commutator to Figure 1a, and specify the impact of this technology on enabling long recording durations in the associated caption.

2. If I read Suppl. Fig. 8 right, the main text should not claim 0.02 mm resolution for “6-degrees-of-freedom head pose”. The maximum error is in z direction and seems more in the range of 2mm.

We have added language that makes it clear that the sub-mm resolution was only achieved in the lateral direction at a 2 meter distance, and that higher resolution can be achieved at smaller tracking distances.

3. An estimate of the orientation accuracy should be given. Many publications like <https://doi.org/10.3390/s20143824> and other AHRS evaluations suggest that the error from sensor fusion of IMUs is substantial in dynamic environments, and the BNO055 will likely not exceed established algorithms. Since 5ocalization is not the topic of this manuscript, a realistic estimation would not impair the impact of ONIX, but it would help experimenters to know what they can expect from a setup like the one presented.

We thank the reviewer for the suggestion. Our initial accuracy estimate was derived from a best-case scenario in which optical tracking as well as optimal the BNO data are available. We have now added some basic quantifications of the orientation accuracy achievable from raw BNO data (quaternions or euler angles as saved by our software) without post-processing and under non-optimal conditions in a new analysis in Supplementary Figure 8:

We performed this measurement with no prior calibration procedure of the BNO and in a non-optimized lab environment with no magnetic shielding. The ~ 2 degrees error range is consistent with the reported accuracy that Bosch states for the BNO055. We now report the correct accuracy figures in the supplement and main text.

As an additional note, the accuracy of the system with regards to small motion where absolute angle is not a major concern can be high enough to discern the sub-degree breathing pattern of a sleeping mouse from head motion, as we now showcase in a new Supplementary Figure 10.

4. Why could twisted pairs used with Neuropixels not be equipped with zero-torque commutation?

In principle this is possible, but prior to the work described in this paper, there were major unresolved engineering challenges: a reliable commutator that can handle GHz frequency data links over many hours without interruptions, a method for measuring animal rotations, and the required control software.

The minimal modifications required to apply our method into the Neuropixels systems would be: Include a BNO or similar sensor on the headstage (inducing some general redesign of the headstage), either adapt the ONI standard or find a way to send the data alongside the neural data in some side-channel, modify our commutator to accept the twisted pair cable with its omnetics connectors, impedance match to the twisted pair instead of our 50Ohm coax, and finally add the required data handling and control algorithms to the firmware and Neuropixels software (SpikeGLX or Open Ephys GUI). This would amount to either using or re-creating significant aspects of our present system.

We met with the Neuropixels team at IMEC at the SfN 2023 meeting and they indicated interest in following this plan, making their system compatible with a modified variant of the commutator described in this manuscript.

5. Suppl. fig. 8: "f-g" legend should be "f-h", or better "f,g,h"?

We have fixed this mistake.

An open tethered bidirectional transmission system that can be flexibly adapted to a variety of sensor platform is certainly a timely and welcome tool for the important and growing field of behavioral quantification of freely moving animal. However a different focus on the novel elements might be advised.

We thank the reviewer for their constructive feedback, our revised paper now addresses this concern by emphasizing the key technological advances and how they make a new type of experiment possible.